# Transcriptome-wide and stratified genomic structural equation modeling identify neurobiological pathways shared across diverse cognitive traits

Andrew D. Grotzinger [1,2] ✉, Javier de la Fuente[3,4], Gail Davies [5,6], Michel G. Nivard [7] & Elliot M. Tucker-Drob [3,4]

Functional genomic methods are needed that consider multiple genetically correlated traits. Here we develop and validate Transcriptome-wide Structural Equation Modeling (T-SEM), a multivariate method for studying the effects of tissue-specific gene expression across genetically overlapping traits. T-SEM allows for modeling effects on broad dimensions spanning constellations of traits, while safeguarding against false positives that can arise when effects of gene expression are specific to a subset of traits. We apply T-SEM to investigate the biological mechanisms shared across seven distinct cognitive traits ($N = 11,263$–$331,679$), as indexed by a general dimension of genetic sharing (g). We identify 184 genes whose tissue-specific expression is associated with $g$, including 10 genes not identified in univariate analysis for the individual cognitive traits for any tissue type, and three genes whose expression explained a significant portion of the genetic sharing across $g$ and different subclusters of psychiatric disorders. We go on to apply Stratified Genomic SEM to identify enrichment for $g$ within 28 functional categories. This includes categories indexing the intersection of protein-truncating variant intolerant (PI) genes and specific neuronal cell types, which we also find to be enriched for the genetic covariance between $g$ and a psychotic disorders factor.

The finding that many diverse cognitive functions are positively intercorrelated was discovered by Spearman in 1904 and has come to be known as one of the most replicated results in psychology[1]. Spearman speculated that the positive manifold of test intercorrelations resulted from their mutual reliance on a common factor, which he termed general intelligence, or $g$, but that each cognitive test also relied on more-specific factors, $s$. Spearman developed factor analysis in order to estimate the relative contributions of $g$ and $s$ factors to a given test from empirical data. Current evidence indicates that $g$

accounts for between approximately 40% and 60%[2] of variation in cognitive test scores. In addition, the $g$-factor is associated with a range of important life outcomes including income level[3], educational attainment[4], social mobility[5], health[6], and longevity[7]. However, despite over 100 years of debate regarding the nature and mechanisms of $g$, the biological mechanisms shared across cognitive functions have remained relatively elusive.

Family-based designs have long indicated that genetic sharing across cognitive functions may underly Spearman's positive manifold[8].

[1]Institute for Behavioral Genetics, University of Colorado at Boulder, Boulder, CO, USA. [2]Department of Psychology and Neuroscience, University of Colorado at Boulder, Boulder, CO, USA. [3]Department of Psychology, University of Texas at Austin, Austin, TX, USA. [4]Population Research Center, University of Texas at Austin, Austin, TX, USA. [5]Lothian Birth Cohorts, University of Edinburgh, Edinburgh, UK. [6]Department of Psychology, University of Edinburgh, Edinburgh, UK. [7]Department of Biological Psychology, VU University Amsterdam, Amsterdam, the Netherlands. ✉e-mail: Andrew.Grotzinger@colorado.edu

More recently, genome-wide association studies (GWAS) have identified specific genetic variants that are associated with the genetic overlap across diverse cognitive traits[9]. A clear next step in this line of research is to characterize the biological pathways implied by these more recent GWAS results. Functional genomic approaches parsimoniously distill the genetic signal across millions of genetic variants into biologically meaningful mechanisms. For example, a recent study of educational attainment (EA) determined that associated genetic variants were enriched for genes involved in specific neurophysiological functions, including synaptic plasticity, ion channel activation, and neurotransmitter secretion[10]. However, existing functional genomic approaches have been developed for univariate applications and, as we demonstrate via simulation, are ill-equipped to analyze multivariate genomic data without false positive inference.

The current study performs a multivariate functional genomic analysis of *g* to both leverage the shared power across seven cognitive traits for discovery and elucidate the biological pathways unique to, and shared across these traits. We specifically apply two multivariate methods. First, we introduce and validate transcriptome-wide structural equation modeling (T-SEM), a method that extends transcriptome-wide association studies (TWAS) approaches to estimate the effects of tissue-specific gene expression within a multivariate system of GWAS traits. Using data from UK Biobank, we apply T-SEM to estimate relationships between gene expression and *g* in order to identify biological mechanisms of sharing across seven cognitive traits. We validate and employ a heterogeneity statistic ($Q_{Gene}$) within T-SEM that quantifies the extent to which the data deviate from the hypothesis that gene expression affects the traits strictly via a common factor, such as *g*. This allows us to identify tissue-specific patterns of gene expression that are associated with only a subset of cognitive traits, or one cognitive trait, such as reaction time. To understand broader biological pathways that transcend the expression of individual genes, we go on to apply another recently developed multivariate functional method, Stratified Genomic SEM[11], to examine genetic sharing and uniqueness within different classes of genetic variants (e.g., variants associated with specific neuronal subtypes). T-SEM and Stratified Genomic SEM are distinguishable with respect to the biological substrate being examined—tissue-specific gene expression versus categories of genes, respectively—but are both applied here with the shared end goal of elucidating the biology that is common and unique across cognitive domains.

## Results

### Overview of T-SEM

T-SEM is a method for examining the effect of tissue-specific gene expression on any parameter within the general Genomic Structural Equation Modeling (Genomic SEM) framework[12]. T-SEM follows a two-stage approach. In Stage 1, univariate, summary-based TWAS is used to perform summary-based transcriptomic imputation (TI) of tissue-specific gene expression on the individual GWAS phenotypes to be included in the model. The analytic pipeline outlined here, and the corresponding open-source publicly available software, specifically utilizes summary-based TWAS output from the FUSION software[13]. Summary-based TWAS is estimated in FUSION as a weighted linear combination of GWAS *Z*-statistics using what are referred to as functional weights. These functional weights are typically pre-compiled from smaller reference datasets containing both tissue-specific gene expression and genotype data and can generally be described as indexing the association between individual single nucleotide polymorphisms (SNPs) and gene expression. Given the costly and intensive nature of obtaining gene expression data, particularly from tissue types such as specific brain regions, summary-based TWAS then allows for drawing inferences about patterns of gene expression associated with complex traits for which only GWAS summary statistics are available. Summary data from each univariate TWAS produced by

FUSION are then combined with one another and with the empirical genetic covariance matrix for the GWAS phenotypes produced using the multivariable version of LDSC[14] within Genomic SEM to create a complete genetic covariance matrix for imputed expression of each gene and the GWAS phenotypes ($S_{Full}$)[12]. An associated sampling covariance ($V_{SFull}$) matrix is also constructed. $V_{SFull}$ includes squared standard errors (SEs) on the diagonal, and sampling covariances on the off-diagonal that quantify dependencies between sampling errors of the estimates. These off-diagonal elements, which are empirically estimated as part of multivariable LDSC, allow T-SEM to be performed for traits with unknown levels of participant overlap across the contributing GWAS.

In Stage 2, the user specifies an SEM in which gene expression is associated with the multivariate system of heritable phenotypes via regression or covariance relationships with components of the model. In our empirical application, the SEM consists of a general factor indexing genetic overlap across seven cognitive traits. We also produce a $Q_{Gene}$ statistic that indexes the extent to which there is a violation of the null hypothesis that imputed expression of a given gene affects the individual traits strictly via the factor. Larger $Q_{Gene}$ statistics occur when gene expression is highly specific to an individual trait or when the gene expression effect is directionally opposing across traits. Thus, we use T-SEM to both identify genes whose tissue-specific expression has general effects on a system of diverse cognitive traits and distill them from genes with more trait-specific effects.

### Validation of T-SEM via simulation

We began by running two sets of simulations to validate the calibration of T-SEM. The first set of simulations generated patterns of gene expression across seven, population-generating conditions. For each condition, separate datasets were simulated for both a top hit and for a gene from the 50th percentile of the *p*-value distribution in our empirical analysis of the *g*-factor. All results reported below follow the same pattern across conditions for both sets of population gene expression effects, apart from the expected decrease in signal for the 50th percentile gene relative to the top hit (see the "Methods" section; Supplementary Figs. 2–5; Supplementary Data 2). The population-generating parameters for the first condition reflected those implied by a model of gene expression operating entirely through a common factor model of genetic *g*. This then reflects a scenario in which the power to detect gene effects on the factor is expected to be high and the signal for $Q_{Gene}$ is expected to conform to a null distribution. Indeed, this is what we observe (Supplementary Data 2), indicating power for discovery and appropriate Type I Error control. The remaining conditions were specified such that the pattern of effects for a gene on the given traits in the population increasingly deviated from the expectations of the common factor model. Results confirmed that as the simulated effect shifts away from the expectation under a common factor model, the power to detect gene expression effects on a common factor and for $Q_{Gene}$ decreased and increased, respectively. Simulation results further revealed that T-SEM associations were not merely a recapitulation of the associations for the most well-powered univariate trait that loads on the factor, that there is a well-controlled false positive rate (FPR) of <5% at $\alpha = 0.05$ when the gene expression effects on the traits is 0 in the population, and that power to detect effects for $Q_{Gene}$ is greatest when the gene effects on the individual traits reflect a mixture of heterogeneous associations that deviate from the expectations of the factor model.

We went on to perform a second set of simulations designed to compare the performance of TWAS of summary statistics from multivariate GWAS (conducted within Genomic SEM) relative to results obtained using T-SEM. These simulations began by specifying population effects at the SNP level that were weighted by the functional weights from FUSION (see the "Methods" section for details). Simulations were again conducted for scenarios that varied in the degree to

which the population generating TWAS effects were reflective of a model in which gene expression operates entirely via the common factor. When the population generating effects were consistent with the expectations implied by the common factor model, we observe 100% power at a Bonferroni-corrected threshold for both TWAS of the common factor summary statistics and T-SEM of FUSION output (Supplementary Data 3). $Q_{Gene}$ was also well-calibrated in this scenario with a 5% FPR at $\alpha = 0.05$. When the population SNP effects were set to 0, we similarly observe a well-controlled FPR, with 4% and 3% of runs significant at $p < 0.05$ for TWAS and T-SEM, respectively, and 7% significant for $Q_{Gene}$. Finally, we find that when the population generating TWAS effects strongly deviates from the factor model, 100% of simulations for both TWAS and T-SEM were significant at a Bonferroni-corrected threshold. However, $Q_{Gene}$ estimated within the T-SEM framework also showed 100% power, thereby safeguarding against the false inference that the effects of gene expression operate at the level of the factor underlying the individual GWAS traits, and appropriately identifying gene expression patterns responsible for trait differentiation. Indeed, TWAS of the common factor summary statistics and T-SEM of FUSION output displayed strong concordance in estimates for the different population generating scenarios (Supplementary Fig. 6 for scatter plots; Supplementary Fig. 7 for QQ-plots), but only T-SEM was able to produce the $Q_{Gene}$ statistic necessary to safeguard against false inference and identify trait-specific pathways of TWAS effects.

## T-SEM analysis of cognitive traits

We applied T-SEM to GWAS summary statistics for seven cognitive traits (Supplementary Data 1) from UK Biobank (UKB): trail-making tests-B, tower rearranging, verbal numerical reasoning (VNR), symbol digit substitution, memory pairs-matching test, matrix pattern recognition, and reaction time (RT). We employ the same common factor model reported in de la Fuente et al.[9], who identified a general dimension of shared genetic liability in these same cognitive traits, which they termed genetic $g$ (Supplementary Fig. 1). These summary statistics were paired with cis gene expression quantitative trait locus (cis-eQTL) reference panel data for 13 brain-based tissue types from GTEx[15] and the two dorsolateral prefrontal cortex (dlPFC) datasets from the Common Mind Consortium (CMC)[16] to produce univariate TWAS estimates in FUSION[13] for 52,849 genes across tissues, which were then used as input for T-SEM.

T-SEM analyses revealed 184 hits for tissue-specific gene expression associated with $g$ that were significant at a Bonferroni corrected threshold, and not significant for $Q_{Gene}$, which explained, on average, 0.13% (range = 0.11–0.25%) of the total genetic variance in $g$ (Fig. 1; Table 1; Supplementary Data 4). As many genes are expressed across multiple reference tissues, these 184 hits ultimately reflect 76 unique genes, including 10 genes that were not significant for any of the univariate TWAS analyses in any tissue type. Gene co-expression analyses using these 76 unique gene IDs as input revealed a total of 59 significant gene sets across three primary clusters (Supplementary Data 6; Supplementary Fig. 10). This included several gene sets implicated in transfer RNA (tRNA), neuron-specific chromatin regulatory BAF subunits (nBAFs), and G-protein-coupled receptors.

Using these 184 T-SEM hits for $g$ that were independent of $Q_{Gene}$ hits as input, joint analyses revealed 29 genes across 18 loci that remained significant when conditioning on shared signal across the hits. These 18 loci explained, on average, 81.1% of the variance of nearby GWAS estimates for $g$ (range = 58–100%; Supplementary Data 5; Supplementary Fig. 9 for regional association plots). In addition, the conditional significance of the top SNP within these loci dropped substantially from an average $p$-value of 1.84E−6 to 0.153, with none of the previous top SNPs within these loci remaining nominally significant. This indicates that inferred gene expression patterns generally account for a large portion of nearby GWAS effects on $g$. As can be visually observed in the Miami plot in Fig. 1, there were sets of $g$ hits

that were physically proximal to genes significant for both $g$ and $Q_{Gene}$ (e.g., on chromosome 3). With this in mind, we went on to rerun joint analyses using the full set of 218 hits, including 34 additional genes significant for $g$ and $Q_{Gene}$. We find that 20 of the 29 genes that were significant from the 184 hit joint analyses remained significant (Supplementary Data 5). Bayesian colocalization analyses were additionally used to examine shared causal variants between hits for gene expression and the $g$-factor, multivariate GWAS (see the "Methods" section). These results revealed that a majority (58.2%) of the 184 $g$ T-SEM hits were supported by a model of colocalized gene expression and GWAS associations (mean posterior probability = 0.568; Supplementary Data 4), with a smaller subset (19.6%; mean posterior probability = 0.223) likely characterized by independent associations. Conservative permutation tests were also used to produce empirically derived univariate TWAS $p$-values used as input for T-SEM (Supplementary Data 4; see the "Methods" section). As is expected for the permutation test, these results indicated an overall attenuation in signal, but largely supported the current findings.

We went on to examine whether the three top hits for $g$ (ZSCAN9, PRSS16, ZNF184) explained a significant proportion of the genetic overlap across $g$ and its clinically relevant correlates. We focus here on findings for an Internalizing disorders factor defined by GWAS summary statistics from major depressive disorder and anxiety disorders[17] and a Psychotic disorders factor defined by bipolar disorder[18] and schizophrenia[19] (see the "Methods" section for details and results for additional traits). We confirm first that genetic $g$ is significantly, genetically correlated with both the Internalizing disorders factor ($r_g = −0.17$, SE = 0.03, $p = 1.33E−8$) and Psychotic disorders factor ($r_g = −0.40$, SE = 0.03, $p = 6.45E−51$), that the three top gene expression hits are significantly associated with the individual traits defining these psychiatric factors (Supplementary Data 7), and that these are not $Q_{Gene}$ hits for the psychiatric factors (Supplementary Data 8). Finally, we find that all three hits explained a significant proportion of the genetic overlap across $g$ and these factors (Supplementary Data 8), with the largest effect observed for ZSCAN9 for both the Internalizing (% mediated $r_g = 1.39$%, SE = 0.21, $p = 3.25E−11$) and Psychotic disorders factor (% mediated $r_g = 1.14$%, SE = 0.14, $p = 7.39E−16$).

For $Q_{Gene}$, we identified 156 hits reflecting 62 unique genes (Supplementary Data 9). These hits reflect genes whose inferred expression is associated with the cognitive traits according to patterns that are inconsistent with their operation on genetic $g$. Similar to $g$-factor hits, Bayesian colocalization analyses supported a model of colocalized associations for the majority of $Q_{Gene}$ hits (51.3%; mean posterior probability = 0.506), while 22.4% of hits likely indexed independent functional and GWAS hits (mean posterior probability = 0.242). Among the most significant $Q_{Gene}$ hits were four unique genes (NSFP1, NSF, ARL17B, LRRC37A) in the 17q21.31 locus. This collection of $Q_{Gene}$ hits consistently evinced a much stronger association with RT relative to the remaining, six cognitive phenotypes (Supplementary Data 9; Supplementary Fig. 12), indicating this region is more relevant to cognitive speed than overall genetic $g$. Within particular tissues, we observed a higher relative mean $\chi^2$ and density of hits for the $g$-factor relative to $Q_{Gene}$, including in the frontal cortex and hippocampus (Supplementary Data 10). Consistent with conceptualizations of the cerebellum as particularly relevant to broad cognitive function[20], this specific tissue showed the largest $g$-factor to $Q_{Gene}$ $\chi^2$ ratio (Supplementary Fig. 13; Supplementary Data 10).

We went on to compare results obtained from T-SEM to results for a TWAS of the $g$-factor GWAS summary statistics. Consistent with simulation results, TWAS and T-SEM estimates for $g$ were highly correlated ($r > 0.99$; Supplementary Fig. 14). For the TWAS of the $g$-factor GWAS summary statistics, we then employed a $Q_{SNP}$ filtering procedure of removing any gene that had functional weights for a $Q_{SNP}$ variant, where $Q_{SNP}$ variants were themselves defined using the T-SEM Bonferroni corrected $p$-value threshold (as opposed to the more

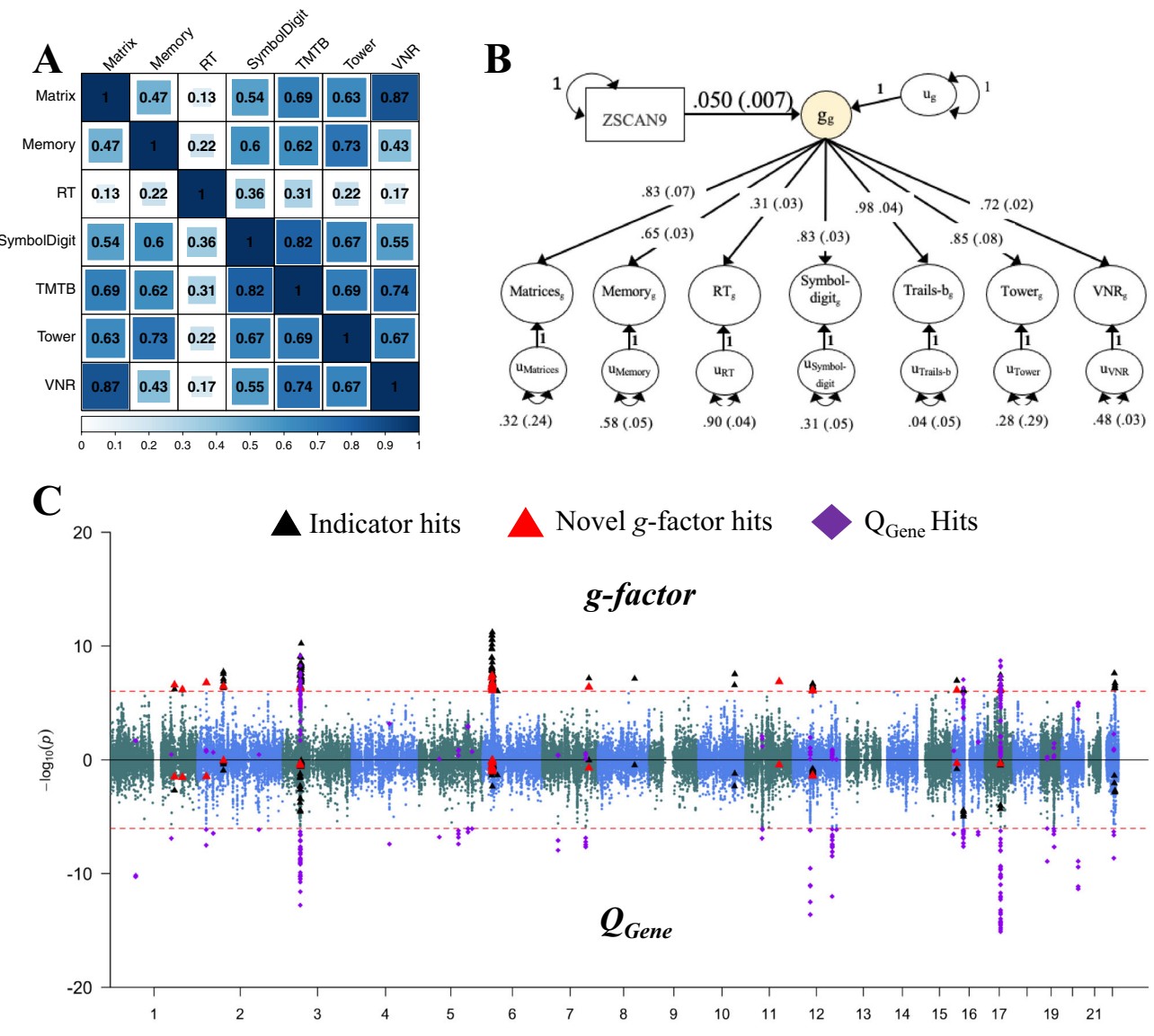

**Fig. 1 | g-factor T-SEM.** *Panel* **A** depicts the genetic correlation matrix as estimated using LDSC for the seven, *g*-factor cognitive phenotypes. The size of the blue-colored square is scaled according to the size of the genetic correlation point estimate. *Panel* **B** depicts the results from the inferred tissue-specific expression of the top gene, *ZSCAN9* in the cerebellum, predicting the *g*-factor estimated using T-SEM. Results are standardized with respect to the genetic variance in the seven cognitive traits, whereas the variance in the genetic *g*-factor is 1+ the variance explained by the individual gene (in this case: $1 + 0.05^2 = 1.0025$). Standard errors are shown in parentheses. *Panel* **C** depicts the Miami plot for *g*-factor T-SEM results.

The top half of the plots depict the $-\log10(p)$ values for TWAS effects on the *g*-factor; the bottom half depicts the $\log10(p)$ values for the *g*-factor $Q_{Gene}$ effects. The red dashed line marks the threshold for TWAS significance using a Bonferroni corrected threshold for 52,849 tests. Black triangles denote *g*-factor TWAS hits that overlapped with hits from univariate analyses for one of the individual cognitive traits. Larger red triangles denote TWAS *g*-factor hits that did not overlap with the hits from univariate analyses of the individual cognitive traits or with hits for $Q_{Gene}$. Purple diamonds denote $Q_{Gene}$ hits.

stringent genome-wide significance threshold). Both of these decisions err on the lax side of excluding more $Q_{SNP}$ signals so as to prevent such signals from contaminating the subsequent TWAS. This $Q_{SNP}$ filtering process failed to remove 23 of the 156 $Q_{Gene}$ hits identified by T-SEM. Moreover, 2 of the 23 remaining $Q_{Gene}$ hits were identified as TWAS hits for *g* and, given significant $Q_{Gene}$ findings, likely to be false positives (Supplementary Fig. 14). We note that in an alternative filtering process of removing $Q_{SNP}$ variants from the *g*-factor, GWAS summary statistics used as input to FUSION failed to remove any of the 156 $Q_{Gene}$ hits. This, in part, reflects the fact that FUSION performs the imputation of missing GWAS *Z*-statistics when possible. Thus, even under conditions selected to ensure that heterogeneous SNP effects were removed, TWAS of the *g*-factor GWAS summary statistics was less

effective than T-SEM at pruning out heterogeneous signals. In summary, findings from both simulations and application to real data indicate that T-SEM is uniquely suited to guard against false positives for effects of gene expression on general factors and to identify patterns of gene expression that underlie genetic divergence among genetically correlated phenotypes.

**Stratified genomic SEM analysis of cognitive traits**
Stratified genomic SEM is a recently developed, multivariate corollary of stratified LDSC[20,21] that can be used to examine functional enrichment of any model parameter estimated in Genomic SEM. Functional enrichment is examined across different classes of genetic variants, referred to as functional annotations, that are grouped according to

**Table 1 | T-SEM results**

| TWAS target | All tissues | | | | Unique gene IDs | | |
|---|---|---|---|---|---|---|---|
| | Mean $\chi^2(1)$ | Hits | Shared hits | Unique hits | Hits | Shared hits | Unique hits |
| $g$-factor | 2.13 | 184 | 150 | 34 | 76 | 66 | 10 |
| $Q_{Gene}$ | 1.91 | 156 | 133 | 23 | 62 | 41 | 21 |
| Matrix | 1.09 | 1 | 0 (0) | 1 (1) | 1 | 0 (0) | 1 (1) |
| Memory | 1.50 | 17 | 1 (0) | 16 (17) | 12 | 1 (0) | 11 (12) |
| Reaction time | 1.95 | 119 | 3 (70) | 116 (49) | 49 | 2 (20) | 47 (29) |
| Symbol Digit | 1.39 | 3 | 2 (0) | 1 (3) | 3 | 2 (0) | 1 (3) |
| TMTB | 1.46 | 25 | 19 (3) | 6 (22) | 16 | 14 (2) | 2 (14) |
| Tower | 1.05 | 0 | 0 (0) | 0 (0) | 0 | 0 (0) | 0 (0) |
| VNR | 2.56 | 507 | 144 (67) | 363 (440) | 183 | 64 (24) | 119 (159) |

For the $g$-factor and $Q_{Gene}$ the *Shared Hits* column reports the number of hits that were overlapping with univariate TWAS hits, while the *Unique Hits* column reports the number of hits that were not identified by univariate TWAS. Total hits for the $g$-factor are reported for Bonferroni significant genes that were not significant for $Q_{Gene}$. $Q_{Gene}$ indexes whether a particular gene is unlikely to operate through the identified $g$-factor structure, as will often be the case when a gene effect is highly specific to an individual trait. For the seven cognitive indicators, the *Shared Hits* column reports univariate TWAS hits that were overlapping with the 184 $g$-factor hits along with values in parentheses reporting univariate hits overlapping with the 156 $Q_{Gene}$ hits. The *Unique Hits* column then reports those univariate hits that were not overlapping with $g$, along with hits unique of $Q_{Gene}$ again reported in parentheses. To facilitate comparison across TWAS targets, mean $\chi^2$ values reported in each row were converted to $\chi^2(1)$ statistics before taking their means. Unique gene IDs were defined as those genes that were significant across any tissue.
*TMTB* trail making test-b, *VNR* verbal numerical reasoning.

some shared characteristic. These shared characteristics can include, for example, whether the variants tend to be conserved in mammals, are associated with specific histone marks, or are implicated in neuronal subtypes. A functional annotation is considered to be enriched, indicating that it is of particular relevance for a given trait, when the genetic variance within that annotation is greater than the proportional size of that annotation. The proportional size of the annotation reflects the number of SNPs in the annotation over the total number of SNPs analyzed. By examining the enrichment of model parameters within a multivariate system of genetically correlated traits, Stratified Genomic SEM facilitates identifying categories for which pleiotropic variation, as separable from trait-specific genetic variation, is enriched.

Using QC and analytic procedures outlined in the "Methods" section, enrichment analyses were based on 155 binary annotations. We observed 28 annotations that were significant for $g$ at a Bonferroni-corrected threshold for 155 tests (Fig. 2; Supplementary Data 11; Supplementary Fig. 16; see Supplementary Method and Supplementary Fig. 17 for enrichment results in standardized space). This included conserved regions, the H3K9ac promoter, and the H3K27ac promoter across different brain regions (e.g., dlPFC, middle hippocampus). Of these 28 significant annotations, four reflected the intersection of PI genes and the GABAergic and excitatory hippocampal and prefrontal cortex neuronal subtypes (Fig. 2).

We observed several significant enrichment estimates for the residual variance components (Supplementary Data 11; Supplementary Figs. 18 and 19), which reflect enrichment of trait-specific genetic variation as separable from genetic $g$. As might be expected, significant estimates were identified for the three cognitive traits with the smallest factor loadings, with 38 significant estimates observed for RT, 33 for VNR, five for the memory pairs matching test, and no significant estimates for the remaining four traits. Among these significant residual estimates, it is perhaps most interesting to consider those that also evinced a weak signal for $g$. This included enrichment for the residuals of VNR and RT for the H3K9ac promoter in the dlPFC, which has been previously associated with Alzheimer's disease[22], and enrichment of the FANTOM5 enhancer for memory-pairs matching.

The PI enrichment findings for $g$ were notably similar to a pattern of enrichment recently described for a psychotic disorders factor defined by bipolar disorder and schizophrenia. As we also observed the previously noted, sizeable, negative genetic correlation between the $g$-factor and a psychotic disorders factor, we went on to examine enrichment of the genetic sharing between these factors (Supplementary Data 11). Results revealed significant enrichment of the factor covariance for 15 annotations, eight of which were also significantly enriched for $g$. The three top annotations enriched for overlap across $g$ and the psychotic disorders factor were PI genes, excitatory prefrontal cortex neurons, and their intersection (Supplementary Fig. 20).

## Discussion

Cognitive functions are characterized by positive intercorrelations at both the observed and genetic levels of analysis. We have used multivariate functional genomic methods to identify both general and trait-specific biological mechanisms of variation across seven cognitive functions. Using T-SEM, we identified 76 unique genes whose inferred expression acts generally across all seven cognitive traits. Highlighting the ability of multivariate methods to leverage shared power for discovery, this included 10 genes that were not significant for univariate TWAS of any of the individual cognitive traits. By aggregating across the understood mechanistic functions of patterns of gene expression associated with $g$, we identified gene sets implicated in transfer RNA (tRNA), neuron-specific chromatin regulatory BAF subunits (nBAFs), and G-protein-coupled receptors. All of these functions have been previously associated with general neural development, cognitive function, and neurodevelopmental disorder risk[23–25]. $Q_{Gene}$, a measure of heterogeneity, identified an additional 62 unique genes whose expression affects individual cognitive traits not via $g$. This included a set of genes on locus 17q21.31 that appear highly specific to cognitive speed. This specific locus is known to be highly pleiotropic and has been linked to frontotemporal dementia[26] and autism[27].

Applying stratified genomic SEM, we additionally identified 28 annotations significantly enriched for $g$. In line with prior univariate functional findings for intelligence[28] and cognitive function[29], we observe significant enrichment in genetic sharing within conserved regions and the H3K9ac promoter. This indicates that these previous univariate discoveries pertain generally to the genetic architecture that is shared across multiple domains of cognitive function. We also find that the H3K27ac promoter was enriched for genetic sharing across several brain regions, including the dlPFC and middle hippocampus, and we observe that PI genes and the GABAergic and excitatory hippocampal and prefrontal cortex neuronal subtypes are highly enriched. The absence of significant findings across all neuronal subtypes points to an increasingly specific set of neurobiological mechanisms that may underlly general cognitive functioning. Highlighting the multivariate capabilities of Stratified Genomic SEM, we also identify a number of annotations that are unique to specific cognitive functions, such as the FANTOM5 enhancer for memory pairs-

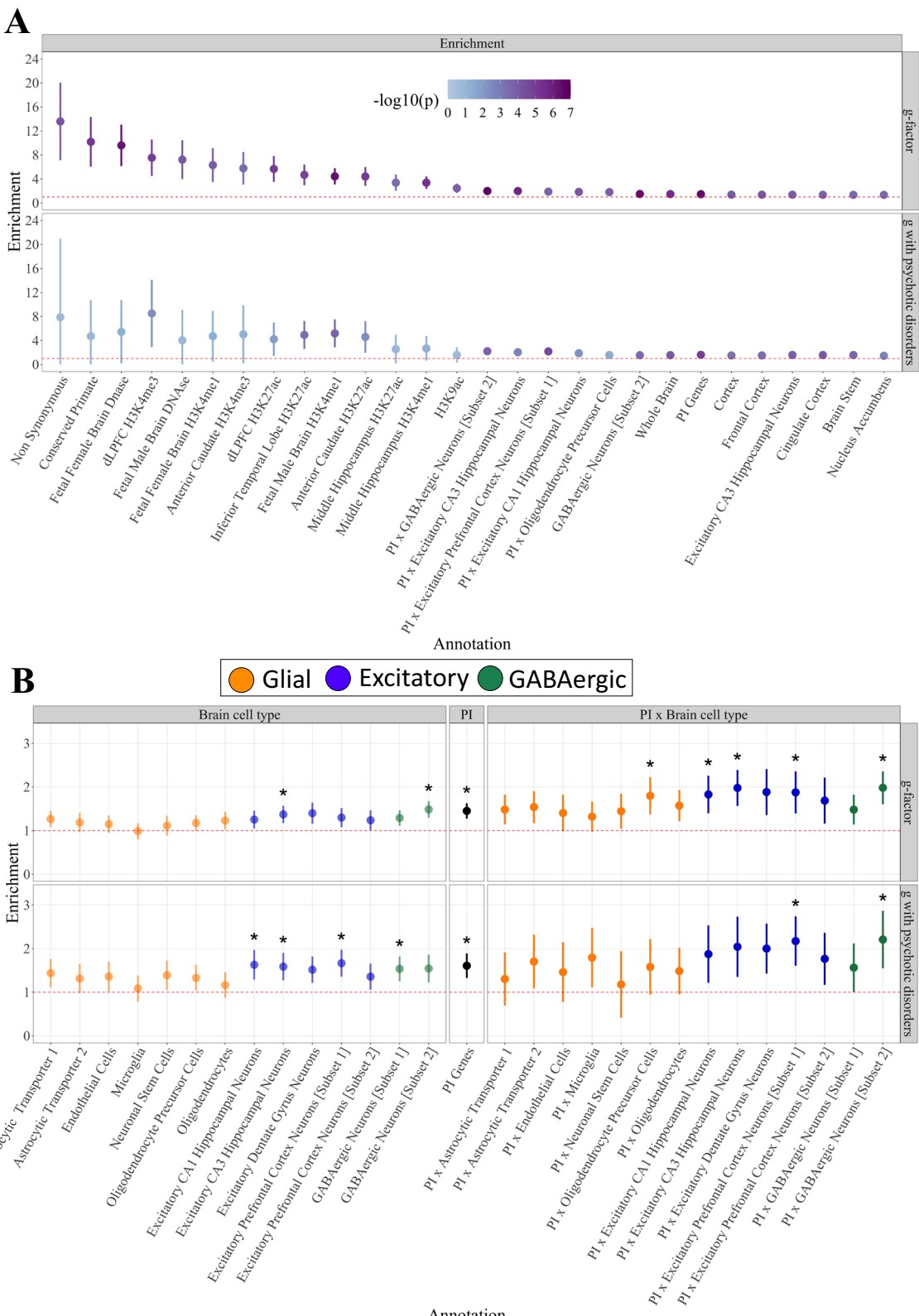

**Fig. 2 | Multivariate enrichment.** *Panel* **A** displays the 28 annotations for the *g*-factor that were significant at Bonferroni corrected threshold for 155 tests. Dots are ordered according to the size of the enrichment point estimate for the *g*-factor and shaded to reflect the level of significance. *Panel* **B** depicts the PI × brain cell annotations and orders the estimates by brain cell type, PI genes, and PI × brain cell type interactions. Glial cells are depicted in orange, excitatory cells in blue, and GABAergic cells in green. Dots that were significant at a Bonferroni-corrected threshold for 155 tests are depicted with a *. In both panels **A** and **B**, the red dashed line reflects the null (enrichment = 1), the dots depict the enrichment point estimates, the error bars depict 95% CIs, and the top half within each panel depicts enrichment estimates for the *g*-factor and the bottom half enrichment estimates for the genetic covariance between the *g*-factor and the psychotic disorders factor.

matching, an annotation that is also strongly linked to immunological diseases[21]. While the exact biological pathways will need to be further elucidated, the identification of the neurobiological building blocks of both $g$ and specific cognitive tests that define $g$ reflects a critical next step in our understanding of this construct.

Major depressive disorder[30], anxiety disorders[31], schizophrenia[32], and bipolar disorder[33] have all been associated with lower cognitive function. In addition, major depressive disorder[34] and schizophrenia[35] have been associated with two of the top $g$ hits from the current analyses: *ZNF184* and *PRSS16*, respectively. As a natural extension of these convergent findings, we find that top gene expression hits on $g$ explained a significant proportion of the genetic overlap across $g$ and these psychiatric traits. Building on patterns of enrichment in neuronal subtypes identified for $g$ that mirrored recently described enrichment across bipolar disorder and schizophrenia, we also find several annotations that are enriched for the genetic covariance between $g$ and these two disorders. This included the intersection between PI genes and both excitatory prefrontal cortex and GABAergic neurons. Collectively, these may represent specific biological pathways underlying the well-established association between cognitive impairment and risk for different clusters of psychiatric conditions.

The positive manifold of intercorrelations across cognitive items lends itself to a factor model of $g$. However, a long-standing source of contention reflects whether $g$ is merely a statistical artifact or, in fact, reflects a useful psychological construct for understanding shared sources of variation across cognitive abilities. Indeed, both Thorndike[36] and Thomson[37] proposed early on that a positive manifold may arise when pairs of cognitive tests share a subset of biological processes (which they termed bonds), even when no biological processes are shared across all cognitive tests. In contrast to this prediction, we identified a number of genes and gene categories that act across all seven cognitive traits under investigation in a manner consistent with their operation via $g$. In contrast, far fewer genes and categories were found to evince patterns of associations inconsistent with their operation via $g$. In combination with recent work identifying 30 individual genetic loci associated with $g$[9], these results lend considerable support to the utility of $g$ at particularly fine (micro) and intermediate (meso) levels of neurobiological resolution.

Future work will benefit from considering developmental aspects of the functional genomics of genetic sharing. For example, processes related to neurobiological organization may be more relevant in early childhood development, whereas processes related to susceptibility vs. resilience to neurodegeneration may be more relevant in late adulthood. As the UK Biobank participants included here reflect individuals ranging in age from approximately 40–75 years of age, future work might particularly focus on early childhood cohorts. Additionally, summary statistics from GWAS of sufficient power to be included in the current analyses were only available for individuals of European ancestry. It will be critical to expand these investigations to more diverse populations.

In summary, we have introduced and validated T-SEM, a method for multivariate TWAS. We applied T-SEM to distinguish genes whose inferred expression operates across seven diverse cognitive functions, as indexed by a genetic $g$-factor, from those whose inferred expression operates more specifically on individual cognitive traits. We went on to apply Stratified Genomic SEM to identify categories of genes relevant to genetic sharing across cognitive traits and others relevant to more specific cognitive traits. We also incorporate psychiatric correlates of $g$ for both types of analysis to identify neurobiological underpinnings shared across cognitive and psychiatric traits. As with our current application to cognitive traits, implementation of the multivariate, functional methods described here can begin to elucidate the biological mechanisms that are shared and unique across genetically correlated quantitative traits and disease phenotypes.

## Methods

### Multivariate TWAS in genomic SEM: T-SEM

Transcriptome-wide structural equation modeling (T-SEM) draws on univariate, summary-based TWAS produced for multiple GWAS phenotypes. In practice, we specifically utilize the FUSION software[13] to perform univariate, summary-based TWAS, which imputes the relationship between gene expression and a trait using the linear combination of GWAS $Z$-statistics and a set of functional weights. For the current analyses, we use the precomputed functional weights available directly from the FUSION website (http://gusevlab.org/projects/fusion/) from cis gene expression quantitative trait locus (cis-eQTL) reference panels (see "Univariate TWAS" section below for details of our specific data sources). These weights are obtained in FUSION by comparing the performance of five different penalized linear models: best linear unbiased predictor (BLUP), Bayesian sparse linear model (BSLMM), elastic-net regression (eNET), lasso regression (LASSO), and single best eQTL (top1)[13]. For each gene, the weights are used from the model that produces the largest $R^2$ between the predicted and observed expression models calculated using five-fold cross-validation.

T-SEM estimation follows the general two-stage approach introduced in the Genomic SEM framework[12], with the goal of modeling genetic covariance between various traits, and the genetically imputed expression level of a gene. In Stage 1 of T-SEM, the genetic covariance matrix and associated sampling covariance matrix across multiple traits are estimated via joint analysis of the univariate GWAS summary statistics for each phenotype in the model using multivariable LDSC. This genetic covariance matrix, which we term $S_{LDSC}$, contains SNP heritabilities on the diagonal and genetic covariances on the off-diagonal. The sampling covariance matrix, which we term $V_{SLDSC}$, is a symmetric matrix composed of the nonredundant elements in the $S_{LDSC}$ matrix. The diagonal elements of $V_{SLDSC}$ are squared SEs of the elements in $S_{LDSC}$. The off-diagonal elements are sampling covariances that index dependencies across estimation errors. These sampling covariances are estimated using a block-jackknife procedure that quantifies the extent to which the sampling distributions of different elements in the $S_{LDSC}$ matrix covary with one another, as would be expected when there is sample overlap across the included traits. The combination of diagonal and off-diagonal elements is what allows Genomic SEM to produce unbiased SEs in the context of the user-specific structural models, even in the presence of unknown levels of sample overlap. Univariate TWAS estimates are subsequently used as input to expand both matrices.

In Stage 2, a model is specified in which gene expression is associated with some other parameter in the model, such as a latent factor defined by the genetic components of the included phenotypes. The model itself can be broken into two parts. The first reflects the measurement model, which parsimoniously describes the genetic relationships across $k$ analyzed traits via a smaller subset of $m$ latent variables. This can be expressed as

$$Y_g = \Lambda\eta + U \qquad (1)$$

where $Y_g$ is a $k$-length vector of the genetic component of the analyzed traits, $\eta$ is an $m$-length vector of latent variables, $\Lambda$ is a $k \times m$ matrix of factor loadings, and $U$ is a $k$-length vector of residual genetic variances not accounted for by the latent variables. The number of latent variables contained within $\eta$, and the patterns of fixed and free parameters contained within $\Lambda$ and $U$ are specified by the user to reflect the model of interest.

In T-SEM, the structural model is then added on top of the genomic measurement model in order to relate tissue-specific gene expression to the latent variables, and the latent variables to one another when >1 latent variable is estimated. The structural model in

T-SEM can be expressed as

$$\boldsymbol{\eta} = \boldsymbol{B\eta} + \boldsymbol{\Gamma}x + \boldsymbol{E}, \tag{2}$$

where $\boldsymbol{\eta}$ is again an $m$-length vector of latent variables, $\boldsymbol{B}$ is an $m \times m$ matrix of regression coefficients that relate latent variables to one another, $\boldsymbol{\Gamma}$ is an $m$-length vector of regression coefficients relating the latent variables to tissue-specific gene expression, $x$ is the tissue-specific gene expression, and $\boldsymbol{E}$ is an $m$-length vector of the residual variances of the latent variables. The terms in $\boldsymbol{B}$, $\boldsymbol{\Gamma}$, and $\boldsymbol{E}$ may include both free parameters and fixed terms, as specified by the user to represent the model of interest. We note that $\boldsymbol{\eta}$ appears on both the left and right sides of the equation as we utilize all-$y$ notation, which does not distinguish between endogenous and exogenous latent variables[38]. The $\boldsymbol{B}$ matrix of regression coefficients then prevents specifying the regression of a latent variable predicting itself by fixing those specific parameters to 0 for that cell of the matrix.

In the context of the current analyses, the $g$-factor measurement model can be written according to the following system of linear equations:

$$
\begin{bmatrix}
\upsilon_{g\text{Matrices}} \\
\upsilon_{g\text{Memory}} \\
\upsilon_{g\text{RT}} \\
\upsilon_{g\text{SD}} \\
\upsilon_{g\text{Trails-B}} \\
\upsilon_{g\text{Tower}} \\
\upsilon_{g\text{VNR}}
\end{bmatrix}
=
\begin{bmatrix}
\lambda_{\text{Matrices}} \\
\lambda_{\text{Memory}} \\
\lambda_{\text{RT}} \\
\lambda_{\text{SD}} \\
\lambda_{\text{Trails-B}} \\
\lambda_{\text{Tower}} \\
\lambda_{\text{VNR}}
\end{bmatrix}
g +
\begin{bmatrix}
u_{\text{Matrices}} \\
u_{\text{Memory}} \\
u_{\text{RT}} \\
u_{\text{SD}} \\
u_{\text{Trails-B}} \\
u_{\text{Tower}} \\
u_{\text{VNR}}
\end{bmatrix}, \tag{3}
$$

where $\upsilon_g$ reflects the genetic component of each of the seven cognitive phenotypes, the $\lambda$'s are the phenotype-specific factor loadings on $g$, and the $u$'s denote the residual genetic variances of the phenotypes. The effect of tissue-specific gene expression on $g$ can then be expressed as

$$g = \gamma x + e, \tag{4}$$

where $\gamma$ is the unstandardized regression coefficient of tissue-specific gene expression on $g$, $x$ is the tissue-specific gene expression, and $e$ is the residual variance of $g$.

To create the $S_{\text{Full}}$ matrix for multivariate TWAS within T-SEM, the $S_{\text{LDSC}}$ matrix is expanded to include the (cis-) genetic covariance between the inferred gene expression and phenotypes, $g_1$ through $g_k$, by appending the vector $S_{\text{Gene}}$:

$$
S_{\text{Full}} =
\begin{bmatrix}
\sigma^2_{\text{Gene}} \\
\sigma_{\text{Gene},g1} & h^2_1 \\
\sigma_{\text{Gene},g2} & \sigma_{g1,g2} & h^2_2 \\
\sigma_{\text{Gene},g3} & \sigma_{g1,g3} & \sigma_{g2,g3} & h^2_3 \\
\vdots & \vdots & \vdots & \vdots & \ddots \\
\sigma_{\text{Gene},gk} & \sigma_{g1,gk} & \sigma_{g2,gk} & \sigma_{g3,gk} & \cdots & h^2_k
\end{bmatrix} \tag{5}
$$

The $\sigma^2_{\text{Gene}}$ in first cell of the matrix above is the (cis-) heritability of the expression of an individual gene, provided directly by FUSION.

The sampling covariance matrix, $V_{\text{SFull}}$, associated with the expanded $S_{\text{Full}}$ covariance matrix, consists of three blocks. The first block is the $V_{\text{SLDSC}}$ matrix obtained from the multivariable LDSC outlined above. The second block, $V_{\text{SGene}}$, is composed of the sampling covariance matrix of the gene expression effects on the phenotypes. The sampling covariances of the gene-genotype covariances with one another are indexed using cross-trait LDSC intercepts. As these cross-trait intercepts reflect sampling correlations (weighted by sample

overlap), they are rescaled relative to the sampling variances of the corresponding gene-genotype covariances to be on the scale of sampling covariances. We treat the sampling variance of $\sigma^2_{\text{Gene}}$ as fixed. In practice, we set it to a very small value (1e−4) and set the sampling covariance between the $\sigma^2_{\text{Gene}}$ and all other terms to 0. The third, and final, block of the $V_{\text{SFull}}$ matrix reflects the sampling covariance of the gene-genotype covariances from FUSION with the genetic variances and genetic covariances from multivariable LDSC. These sampling covariances are fixed to 0 as the gene-phenotype covariance will be independent of the individual SNP effects in all LD blocks except those which SNPs defining the gene occupy. As the sampling variance of the elements of $S_{\text{LDSC}}$ reflects sampling variability with SNP test statistics in all LD blocks, their sampling covariance with the effect of a single gene is expected to approach 0. Taking these three components together, the $V_{\text{SFull}}$ matrix for multivariate TWAS can be written in compact form as

$$
V_{S_{\text{Full}}} =
\begin{bmatrix}
V_{S_{\text{Gene}}} & \\
0 & V_{S_{\text{LDSC}}}
\end{bmatrix} \tag{6}
$$

The $S_{\text{Full}}$ and $V_{\text{SFull}}$ matrices are constructed as many times as there are shared gene expression IDs across univariate FUSION outputs for individual traits.

### Scaling TWAS output for T-SEM

The output from univariate summary-based TWAS (as estimated with FUSION[13]) is a predicted gene-trait $Z$-statistic that requires further transformation to be added to the $S_{\text{Full}}$ matrix above. First, this $Z$-statistic is converted to a partially standardized regression coefficient and its SE. For continuous traits, this follows the equations $b_{\text{Gene,P}} = \frac{Z}{\sqrt{N \sigma^2_{\text{Gene}}}}$ and $SE_{b\text{Gene,P}} = \frac{b_{\text{Gene,P}}}{Z}$, where $\sigma^2_{\text{Gene}}$ reflects the heritability estimate for an individual gene provided by FUSION. We highlight that this metric is somewhat arbitrary as we treat the genetic variance captured by FUSION imputation ($\sigma^2_{\text{Gene}}$) as equivalent to the total cis-genetic variation, whereas in reality these numbers may depart substantially. Importantly, the chosen metric for a gene will have the same effect on the metric of its associations with all phenotypes included in the model, such that the choice of scaling will neither bias the pattern of associations nor affect downstream T-SEM, such as $Q_{\text{Gene}}$ estimation and associated $p$-values. Other sensible choices of scaling include the out-of-sample $R^2$ reported in FUSION. While this choice would change the metric of effect sizes, it will not affect the pattern of results or associated $p$-values. For binary traits, unstandardized logistic regression coefficients and SEs are first backed out from the FUSION $Z$-statistics as $b_{\text{logit}^*\text{Gene,P}} = \frac{Z}{\sqrt{N_{\text{eff}}/4\sigma^2_{\text{Gene}}}}$ and $SE^*_{b\text{logitGene,P}} = \frac{1}{\sqrt{N_{\text{eff}}/4\sigma^2_{\text{Gene}}}}$, where $N_{\text{eff}}$ reflects the sum of effective sample sizes across cohorts that contribute to the GWAS meta-analysis. These logistic regression coefficients and *SEs* are then partially standardized by dividing both by $\sqrt{\sigma^2_{\text{Gene}} \times (b_{\text{logit}^*\text{Gene,P}})^2 + \frac{\pi^2}{3}}$, where $\frac{\pi^2}{3}$ reflects the residual variance from a logistic regression model. We refer to these as partially standardized regression coefficients as they are standardized relative to the variance in the outcome (i.e., the phenotype of interest), but not the predictor (i.e., a given gene). These partially standardized coefficients are subsequently transformed into gene-phenotype covariances ($\sigma_{\text{Gene,P}}$), as in the $S_{\text{Full}}$ matrix above, by multiplying by the variance (heritability) of the gene ($\sigma^2_{\text{Gene}}$.)

### $Q_{\text{Gene}}$ Test of heterogeneity

$Q_{\text{Gene}}$ indexes whether a given gene is more likely to operate through an identified common factor or via the independent pathways of the individual traits that define the factor. This metric helps to guard against identifying genes as operating through the common factor

when they are, in fact, highly specific to a trait or subset of traits. That is, it formally tests the null hypothesis that the gene acts through a given factor. $Q_{Gene}$ is calculated here using a two-step procedure that mirrors the steps outlined for $Q_{SNP}$[9,12]. In Step 1, a common pathway model is fit where the gene effect on the common factor, the (residual) genetic variances of the factor and the phenotypes that define the factor, and all but one-factor loading that is scaled to unity for identification, are freely estimated. In Step 2, a common plus independent pathways model is fit where the factor loadings and the gene effect on the common factor are fixed from the parameter estimates in Step 1, and the direct effects of the gene on the phenotypes and the residual variances are freely estimated. Supplementary Figure 1b depicts this two-step procedure, as applied to the g-factor model, with parameters that are fixed in Step 2 depicted in red and those that are freely estimated in Step 2 depicted in black. The same formula used for estimates of model $\chi^2$ in Genomic SEM is used here to produce a $\chi^2$ distributed $Q_{Gene}$ test statistic with degrees of freedom (df) equal to $k-1$, where $k$ reflects the number of included phenotypes. For comparative purposes throughout the paper, we generally scale this $Q_{Gene}$ test statistic to be a $\chi^2$ statistic with df = 1.

## Simulations of gene expression

**Simulation procedure.** As a first step in the simulation procedure, a model implied genetic covariance matrix was calculated for a model in which gene expression from real data analyses was specified to predict the g-factor. This reflects a best-case scenario for a common factor signal in T-SEM as the model implied matrix reflects a pattern of relationships that operate entirely through the common factor. We utilized two-model implied matrices, one from gene expression results from the top hit (*ZSCAN9* expression in the cerebellum), and the other from results for the gene with null signal reflecting the 50th percentile real data results for the g-factor, T-SEM p-values (*ZNF749* in the hippocampus). Seven versions of these two model implied matrices were subsequently used to construct population-generating covariance matrices, from which 1000 genetic covariance matrices were then sampled using rmvnorm in R for each of the seven scenarios (i.e., 14,000 total simulated datasets). These seven scenarios consisted of: Scenario 1, reflective of the alluded to best-case scenario where the model implied matrix was unchanged; Scenario 2 in which the covariance between the gene and RT (the phenotype with the smallest factor loading) was set to 0; Scenario 3 in which the covariance between the gene and Trails-b (the phenotype with the largest loading) was set to 0; Scenario 4 in which the covariance between the gene and all cognitive phenotypes except trails-B was set to 0; Scenario 5 in which the covariance between the gene and all phenotypes except RT was set to 0; Scenario 6 in which the covariance between the gene and all phenotypes was set to 0; and Scenario 7 in which the directionality of the covariance between the gene and matrices, memory, and RT was reversed.

The observed sampling covariance matrix was used both to sample from the population matrices for each condition and was paired with each simulated genetic covariance matrix for the real data analyses. This has the advantage of providing a set of simulations that are both directly relevant to the interpretation of the current analyses and, by definition, reflective of the types of real-data scenarios where T-SEM might be applied. All simulations applied the same Bonferroni corrected threshold for significance as used in the real-data analysis.

**Simulation results.** These scenarios were selected to reflect a gradient of deviations from the factor model, where Scenario 1 exactly matches the common factor model and Scenario 7 reflects a strong departure from the model wherein gene expression has directionally opposing effects across the g-factor phenotypes. This allowed us to test whether T-SEM appropriately down weights estimates of gene expression

effects on the common factor in a graded fashion as the generating population gradually shifts further away from the common factor structure. Results for Scenario 1, with population gene effects specified to operate solely through the common factor, revealed hits on the common factor for all but 1 run for the *ZSCAN9* (top hit from real data) simulations (i.e., 99.6% hits; Supplementary Data 2) and evinced the strongest signal across scenarios (Supplementary Fig. 4). The signal was slightly reduced for Scenario 2, in which the gene effect was set to 0 for the cognitive phenotype with the smallest loading (RT) and more attenuated for Scenario 3, in which the gene effect was set to 0 for the phenotype with the largest loading (Trails-b; Supplementary Fig. 2). It is also worth noting that Scenario 3 still produced hits for 82.7% of the top hit simulations. As expected, the signal was greatly reduced relative to these first three scenarios for Scenario 4, in which the gene effect was set to 0 for all but Trails-b, and further reduced for Scenario 5, in which the gene effect was set to 0 for all but RT, with no simulations producing hits for either scenario. Taken together, this demonstrates that phenotypes with larger factor loadings are appropriately weighted when estimating the effects of gene expression and that the signal for the common factor is by no means a recapitulation of the signal for the strongest phenotype. For Scenario 6, in which the gene expression effect in the population was 0 across all phenotypes, simulation results revealed a null signal, with a well-controlled FPR at $p < 0.05$ of 4.96% for the top hit simulations and 3.78% for the null signal simulations. Finally, for Scenario 7, in which the directionality of the gene effect was reversed for three of the phenotypes, 35.1% of the runs were identified as significant for the common factor for the top hit simulations. Importantly, all significant runs in this scenario were also identified as hits for $Q_{Gene}$, which we consider next.

The pattern of results of $Q_{Gene}$ was also consistent with expectations. For Scenario 1, there was null signal, with no simulations identified as $Q_{Gene}$ hits, and an FPR at $p < 0.05$ of 3.90% for top hit simulations and 6.70% for the null signal simulations. These Scenario 1 results are consistent with the fact that the population did not deviate from the factor structure (Supplementary Figs. 3 and 5). The next closest signal was for Scenario 6 in which all associations were set to 0 in the population, reflecting the fact that a pattern of null associations across the phenotypes largely aligns with a common factor model for highly correlated traits. Scenarios 2 and 5, in which the population effect of the gene on RT and all phenotypes except RT were set to 0, respectively, evinced a similar $Q_{Gene}$ signal that was slightly stronger than Scenario 6 but did not produce any $Q_{Gene}$ hits. These results reflect the relatively smaller influence of RT (the phenotype with the smallest loading) on the overall factor model. Scenarios 3, in which the population effect of the gene on Trails-B was 0, evinced the next highest signal, with 39.4% of runs producing $Q_{Gene}$ hits for the top hit simulations, followed by Scenario 4, in which the gene effect on all phenotypes except Trails-B was set to 0 in the population, with 46.1% of top hit runs producing $Q_{Gene}$ hits. As expected, Scenario 7, which deviated the strongest from the model with directionally opposing gene effects on matrices, memory, and RT relative to the remaining phenotypes, showed the strongest $Q_{Gene}$ signal by far, with 99.6% of top hit simulations identifying a $Q_{Gene}$ hit. As would be expected, none of the simulations produced factor or $Q_{Gene}$ hits at a Bonferroni corrected threshold for the generating population that reflected the 50th percentile of the real data, g-factor results. However, the patterning of results across conditions mirrored that for the top hit simulation results.

## SNP-level simulations

**Simulation procedure.** Our next simulations sought to compare the performance of TWAS of a common factor, GWAS summary statistics generated using Genomic SEM relative to results obtained using T-SEM of FUSION output for the GWAS phenotypes that define the common factor. These simulations began by generating 100 sets of genetic

covariance and sampling covariance matrices following the procedure described in de la Fuente et al. (2021)[39] Data were simulated using European population LD scores for 1,184,461 HapMap3 SNPs, excluding the MHC region, for five continuous traits. These traits were specified to be 20% heritable in the population, genetically and phenotypically correlate at 0.7, and have minimal population stratification (univariate LDSC intercept = 1.04). All traits had a sample size of 150,000, with no sample overlap across the traits. These GWAS summary statistics were simulated from the multivariable LDSC equation as

$$\left[ Z_{1j}, Z_{2j}, \ldots Z_{5j} \right] \sim N\left( [0,0,\ldots 0], \mathrm{cov}\left( Z_{1j}, Z_{2j}, \ldots Z_{5j} \right) \right) \quad (7)$$

where,

$$\mathrm{cov}(Z_{1j}, Z_{2j}, \ldots Z_{5j}) = \begin{bmatrix} N_1 \frac{h_1^2}{M} \ell(j) + 1 + a_1 & & & \\ \sqrt{N_1 N_2} \frac{\sigma_{g1,2}}{M} \ell(j) + \frac{\rho_{1,2} N_{s1,2}}{\sqrt{N_1 N_2}} & N_2 \frac{h_2^2}{M} \ell(j) + 1 + a_2 & & \\ \vdots & \vdots & \ddots & \\ \sqrt{N_1 N_5} \frac{\sigma_{g1,5}}{M} \ell(j) + \frac{\rho_{1,0} N_{s1,5}}{\sqrt{N_1 N_5}} & \sqrt{N_2 N_5} \frac{\sigma_{g2,5}}{M} \ell(j) + \frac{\rho_{2,5} N_{s2,5}}{\sqrt{N_2 N_5}} & \cdots & N_5 \frac{h_5^2}{M} \ell(j) + 1 + a_5 \end{bmatrix}$$

$$(8)$$

and $\left[ Z_{1j}, Z_{2j}, \ldots Z_{5j} \right]$ reflects the $Z$-statistics for the five GWAS cohorts, $N$ is the sample size of the individual GWAS that was set to 150,000 for all five phenotypes, $N_s$ is the number of overlapping individuals (0 in this case), $M$ is the number of SNPs from the LD file (1,184,461), $\rho$ is the phenotypic correlation (set to 0.7 here, but inconsequential when there is no sample overlap), $\ell(j)$ is the LD score of SNP $j$, and $a + 1$

$$\left[ \widehat{b_{GWAS1}}, \widehat{b_{GWAS2}}, \ldots \widehat{b_{GWAS5}} \right] \sim N$$

$$\left( [b_{GWAS1}, b_{GWAS2}, \ldots b_{GWAS5}], \begin{bmatrix} \frac{1-2MAF(1-MAF)b_{GWAS1}^2}{2MAF(1-MAF)n_1} & & & \\ 0 & \frac{1-2MAF(1-MAF)b_{GWAS2}^2}{2MAF(1-MAF)n_2} & & \\ \vdots & \vdots & \ddots & \\ 0 & 0 & \cdots & \frac{1-2MAF(1-MAF)b_{GWAS5}^2}{2MAF(1-MAF)n_5} \end{bmatrix} \right) \quad (12)$$

reflects the univariate LDSC intercept indexing unmeasured confounds such as population stratification (1.04). The bivariate LDSC intercept, expressed as $\frac{\rho_{1,2} N_{s1,2}}{\sqrt{N_1 N_2}}$ for phenotypes 1 and 2, reduced to 0 for all pairs of phenotypes as the sample overlap was 0. These simulated GWAS summary statistics were then used as input to multivariable LDSC to produce 100 sets of genetic covariance matrices and associated sampling covariance matrices and subsequently paired with simulated SNP effects for the *GNL3* gene for five population-generating scenarios.

The population-level SNP effects on the five phenotypes ($b_{GWAS}$) were computed using three, key population parameters: (i) the SNP effect on gene expression ($b_{eQTL}$), (ii) the effect of gene expression on the common factor ($\gamma$), and (iii) the effect of the common factor on the phenotypes (i.e., the factor loadings; $\lambda_i$). The population SNP effects on gene expression were taken from the pre-compiled weights provided by FUSION. We specifically use the top1 weights as this reflects the best-performing model for *GNL3* in the basal ganglia. The top1 weights refer to the single best eQTL weights, where only the individual SNP with the largest weight is used for TWAS, in this case, rs1108842. The top1 weight is computed by FUSION as:

$$\mathrm{top1} = \frac{GE}{\sqrt{n-1}}, \quad (9)$$

where $G$ is a matrix of standardized genotypes, $E$ is a vector of standardized gene expression from the same participant sample, and $n$ is the number of samples used to calculate the weights, which was 144 for *GNL3* expression in the basal ganglia. For the purposes of the current simulations, the top1 weight was converted to a partially standardized, population $b_{eQTL}$ (i.e., standardized with respect to gene expression, but not standardized with respect to the SNP variance). This was calculated as:

$$\mathrm{be_{QTL}} = \frac{\mathrm{top1}}{\sigma_{SNP} \sqrt{n-1}}, \quad (10)$$

where $\sigma_{SNP}$ is the standard deviation of the SNP calculated from the 1000 Genomes Phase 3 European reference panel as 0.706 for rs1108842. These calculations then yielded a population $b_{eQTL}$ of 0.442. The true effect ($\gamma$) of tissue-specific gene expression on the common factor was set to reflect the beta coefficient (scaled from a model using unit variance identification) from the real data T-SEM results of *GNL3* for the g-factor ($b = -0.089$), as this reflects a realistic point estimate. The unstandardized population effect of the common factor on the phenotypes ($\lambda_i$, also scaled using unit variance identification) was 0.374 using the population parameters described above. Putting these pieces together, $b_{GWAS,i}$ for each trait, $i$, in the population was calculated as

$$b_{GWAS,i} = b_{eQTL} * \gamma * \lambda_i. \quad (11)$$

Observed SNP effects for each phenotype were then sampled from the sampling distribution given as

Note that the off-diagonals of the sampling covariance matrix for the betas were 0, as expected under conditions of 0 sample overlap (otherwise, the off-diagonal elements are determined by the sampling correlation, equal to the cross trait LDSC intercept, as given earlier, rescaled to covariances using the corresponding sampling variances that are on the diagonals). The population betas, and the corresponding sampling distribution, were perturbed from their expectations above to reflect five population-generating scenarios. Scenario 1 reflected one in which gene expression operated entirely through the common factor, such that the population betas were calculated as described (i.e., no perturbation of population betas). Scenario 2 deviated from the factor model where the direction of the population betas was reversed for three of the five phenotypes, and Scenario 3 deviated still further where the direction was reversed, and the population effect doubled for three of the five phenotypes. For Scenario 4 the population betas were set to 0 for three of the five phenotypes, and for Scenario 5 the population betas were set to 0 for all phenotypes. We sampled 100 sets of SNP effects for each of the five population-generating scenarios.

These simulated SNP-phenotype betas and their standard errors were then used in analytic pipelines both for T-SEM and for an ad hoc univariate TWAS of summary statistics from multivariate GWAS. The T-SEM pipeline involved inputting the simulated SNP-phenotype Z-statistics for each phenotype to FUSION, converting these phenotype-

level FUSION estimates to gene-phenotype covariances, and appending these FUSION estimates to the simulated genetic covariance and sampling covariance matrices. These matrices were then used as input to T-SEM to produce both estimates of gene expression on the common factor and the $Q_{Gene}$ heterogeneity estimate. To produce ad hoc TWAS estimates for the common factor, the SNP-phenotype betas were converted to SNP-phenotype covariances so that they could be appended to the 100 simulated genetic covariance and sampling covariance matrices. These expanded matrices were then used as input into Genomic SEM to produce estimates of SNP effects on the common factor, and the common factor GWAS summary statistics were then used as input for univariate analysis in FUSION.

We ran an additional set of simulations that examined the effect of sampling variation in estimates of the SNP effect on gene expression ($b_{eQTL}$). The sampling variance (i.e., the squared standard error) of the $b_{eQTL}$ estimate is specifically given as

$$\sigma^2_{beQTL} = (SE_{beQTL})^2 = \frac{1 - \sigma^2_{SNP}(b^2_{eQTL})}{\sigma^2_{SNP}(n-1)}. \tag{13}$$

For Scenario 1, where the generating population matched the factor model, we then generated a set of 100 population $b_{eQTL}$ estimates using the rnorm function in R. These were subsequently used to create a new set of population $b_{GWAS}$ estimates for each simulation run, the SNP effects ($b_{GWAS}$) sampled from the sampling distribution given above, and the remainder of the simulation pipeline conducted to mirror the other SNP-level simulations. Importantly, the FUSION weight for *GNL3* was left unchanged, such that each simulation varied in the degree of mismatch between the population $b_{eQTL}$ and the functional weight used to produce TWAS estimates.

**Simulation results.** Results for Scenario 1, for which the population generating parameters perfectly matched a model of tissue-specific gene expression operating through the common factor, revealed 100% power for both TWAS of the multivariate GWAS summary statistics for the common factor (TWAS$_{Factor}$) and T-SEM using a Bonferroni corrected threshold of $p < 9.46E{-}7$ and, as would be expected, 0% positivity for $Q_{Gene}$ (Supplementary Data 3). For Scenario 2, characterized by the population direction SNP effect reversed for three of the five phenotypes, there was 0% positivity for TWAS$_{Factor}$ and T-SEM and 100% power for $Q_{Gene}$. Scenario 3 then deviated still further from the factor model with the direction of SNP effects reversed and doubled for three of the five phenotypes. In this instance, the heterogenous population effects were large enough that we observe 100% positivity for TWAS$_{Factor}$ and 97% positivity for T-SEM. Critically, we find also that $Q_{Gene}$ has 100% power. This scenario demonstrates the key advantage of the T-SEM framework relative to performing a TWAS of multivariate GWAS summary statistics produced from multivariate methods like Genomic SEM. That is, while we observe strong concordance between TWAS$_{Factor}$ and T-SEM estimates (Supplementary Data 3; Supplementary Fig. 6), only T-SEM can employ $Q_{Gene}$ as a means of both guarding against false positives and identifying patterns of gene expression that are specific to a trait or subset of traits. For Scenario 4 and 5 simulated population betas of 0 for three and five of the phenotypes, respectively, the signal was highly attenuated and did not evince observable deviations from the null (Supplementary Fig. 7 for QQ-plots). We observe 7% positivity for TWAS$_{Factor}$ and T-SEM for Scenario 4, with 2 of the 7 T-SEM hits identified as significant for $Q_{Gene}$, and 9% power for $Q_{Gene}$. Finally, for Scenario 5 we observe 0% positivity across TWAS$_{Factor}$, T-SEM, and $Q_{Gene}$.

We end by considering the false positive rate (FPR) at $p < 0.05$ for the two scenarios for which the population effects can be clearly considered as reflective of a given null hypothesis. More specifically, Scenario 1 which perfectly matches the common factor model reflects the null for $Q_{Gene}$, and in this case, we observe a 5% FPR. In addition,

Scenario 5 is reflective of the null for both the common factor signal and $Q_{Gene}$ for our specific simulating parameters where the equal factor loadings across the phenotypes mirror a pattern of population betas of 0s across the phenotypes. Once again, we find a well-controlled FPR, with 4% FPR for TWAS$_{Factor}$, 3% for T-SEM, and 7% for $Q_{Gene}$. In summary, SNP-level simulation results indicate that the positivity rate (i.e., power) for TWAS$_{Factor}$ and T-SEM appropriately scales as a function of the size of population effects and the degree to which the effects of gene expression on the individual traits correspond to the expectations of the factor model, that FPR is well controlled for population scenarios that reflect a given null, and that TWAS$_{Factor}$ and T-SEM yield a concordant set of association results, with the critical exception that the $Q_{Gene}$ statistic estimated within T-SEM guards against false positives and identifies sources of differentiation of gene expression effects across traits.

Simulation results that included sampling variation in the $b_{eQTL}$ estimates are visually summarized in Supplementary Fig. 8. We find that, as with the other simulation findings, there was a tight correspondence between T-SEM estimates and those from TWAS$_{Factor}$, that these are both well-powered approaches when the population matches the factor model with 93% of the simulation runs significantly at a Bonferroni corrected threshold, and that $Q_{Gene}$ evinces a well-controlled FPR of 5% at $p < 0.05$. Finally, as would be expected, the downstream consequence of including variation in the population $b_{eQTL}$ estimates was a wider sampling distribution of estimates relative to the simulations that treated $b_{eQTL}$ as a known value (Supplementary Data 3). As with univariate TWAS, functional weights estimated from finite samples will result in greater variation in estimates relative to the population. As the gene expression samples used to train these functional weights increase, imprecision of the TWAS weights will exert an increasingly minimal influence on the precision of downstream estimates.

## Univariate TWAS

The FUSION software[13] was used to perform transcriptomic imputation (TI)/summary-based univariate TWAS for the seven cognitive summary statistics. Functional weights were used from eQTL reference panels for 13 brain tissue panels from the Genotype-Tissue Expression project v7 (GTEx; $n = 753$; https://gtexportal.org/home/datasets)[15] and 2 dlPFC panels from the CommonMind Consortium (CMC; $n = 452$)[16]. The functional weights utilized in the current study are all publicly available on the FUSION website and were pre-computed using the package defaults. These weights were coupled with LD information from the 1000 Genomes v3 European subsample to produce univariate TWAS test statistics. The FUSION package quality control defaults were also used for summary-based TWAS, including a minimum $R^2$ imputation accuracy of 0.7 per gene and a maximum of 50% of SNPs allowed to be missing per gene. Using these defaults, results were not calculated for 233 genes, for a remainder of 52,849 genes across the 15 tissues. A strict Bonferroni-corrected threshold was used for both the g-factor T-SEM results and $Q_{Gene}$ test statistics using an FDR of 0.05 (0.05/52,849).

TWAS test statistics produced by FUSION are well-calibrated under the null of no GWAS association but can become inflated as a result of random quantitative trait loci (QTL) colocalization. This can occur when a locus is both highly significant and characterized by extensive LD. To guard against these instances, FUSION offers a permutation test statistic that recomputes the TWAS test statistic conditional on the GWAS effects at that locus after randomly reordering the QTL weights. This permutation test asks whether the distribution of QTL effect sizes is *by itself* sufficient for producing a significant TWAS association. The output is an empirically derived p-value that indexes the proportion of permutations that were more significant than the observed TWAS p-value. For the current analyses, 100,000 permutations were run per gene for all genes. These univariate, empirical

*p*-values were then used as input for a separate, multivariate TWAS of the *g*-factor in Genomic SEM. It is important to note that these univariate empirical *p*-values, and consequently the multivariate TWAS results, are highly conservative such that genes that are truly causal in the population may fail to reject the null when their QTLs are characterized by extensive and complex patterns of LD. At the same time, genes that remain significant for the permutation test can be interpreted as less likely to be colocalized due to chance.

### T-SEM follow-up analyses

All follow-up analyses were computed at the level of the *g*-factor, as opposed to at the level of the individual cognitive traits used as input for the multivariate TWAS.

**Conditional analyses.** As many genes are present across tissue types, and genes overlap in physical proximity, it can also be useful to consider the conditional effect of each gene. Conditional analyses are conducted in FUSION via an iterative procedure that adds predictors to the model until no significant associations remain. Analyses were conducted using the package default locus window of 100,000 bp. It is of note that genes that are estimated to be jointly significant are not necessarily causal and those that are not jointly significant may still be causal. This is due to the fact that the gene expression features in the latter case may simply be characterized by high correlations with other features. It is also useful to consider to what extent the model explains observed SNP effects by examining SNP effects conditional on TWAS estimates. In this context, it is informative to examine both the level of significance of the top SNP within a region before and after conditioning on TWAS estimates along with the proportion of variance explained in that region by corresponding TWAS results. If TWAS estimates explain a small proportion of the GWAS variance in a region, this suggests that TWAS estimates are tagging an independent causal feature, with the inverse being true when large amounts of variance are explained.

**Colocalization analyses.** Colocalization analyses can also be used to examine the probability of a shared causal variant between gene expression and the trait of interest (i.e., whether there are colocalized functional and GWAS associations). This reflects an alternative to a TWAS, which examines the evidence of a signification association between imputed gene expression and the trait. Bayesian colocalization analyses were conducted using the coloc R package[40] run through FUSION. When implemented via FUSION, the coloc package works by estimating the posterior probability of different configurations of a single causal SNP for both gene expression and the trait, along with the posterior probability that the gene expression and trait share these configurations. The output is posterior probabilities for five scenarios. Model 0 (PP0 in Supplementary Data 2) reflects a situation in which there is no GWAS or functional association. Model 1 (PP1) examines the probability of a functional association only and Model 2 (PP2) examines the probability of a GWAS association only. Model 3 (PP3) examines whether there are independent functional and GWAS associations. Finally, Model 4 (PP4) examines the probability of colocalized functional and GWAS associations. As the *coloc* software assumes a single causal variant, and FUSION models assume multiple eQTLs, a low posterior probability of Model 3, as opposed to a high posterior probability of Model 4, can also be taken as a good indication of colocalization. These posterior probabilities were calculated using the *g*-factor GWAS summary statistics and functional reference weights across tissues as input.

**Gene-set analyses.** GeneNetwork v2.0[41] was used to estimate gene co-expression networks in order to better characterize the multivariate TWAS results for both the *g*-factor and $Q_{Gene}$. Genes used as input for the *g*-factor to create the co-expression network included those genes

that were significant at a Bonferroni corrected threshold for 52,849 tests and did not overlap with significant $Q_{Gene}$ hits for the same gene and tissue type. These genes were restricted still further to those unique gene IDs across tissue types for a total of 76 genes used as input. There were 62 unique genes for $Q_{Gene}$, among which 3 were not present in the database, for a total of 59 gene IDs used as input. Pathways were subsequently analyzed across 3,033 pathways from the Reactome database and the three primary Gene Ontology (GO) databases for biological processes, molecular functions, and cellular components. A Bonferroni corrected threshold was used to identify significant pathways for 3033 tests at an FDR of 0.05 (i.e., $p < 1.65E{-}5$).

**Gene expression mediation of overlap with clinically relevant correlates.** Both *g* and the top gene expression hits for *g* have been associated with several clinically relevant outcomes. This includes previously described associations for Alzheimer's disease and *ZSCAN9*[42], for major depressive disorder (MDD)[34] and Parkinson's disease (PD)[43] and *ZNF184*, and for schizophrenia (SCZ)[35] and *PRSS16*. Following up on this work, we examined whether these patterns of gene expression explained a significant proportion of the genetic overlap across *g* and these correlates of *g* by utilizing publicly available GWAS summary statistics for ALZ[44] PD[45], MDD[46], and SCZ[19]. In line with the multivariate framework employed here, we examine MDD and SCZ in conjunction with GWAS summary statistics for anxiety disorders (ANX)[17] and bipolar disorder (BIP)[18], respectively, given high levels of genetic overlap across these pairs of disorders. This was done by specifying a two-phenotype Internalizing disorders factor consisting of MDD and ANX and a Psychotic disorders factor consisting of BIP and SCZ; we note that the loadings on these two-phenotype factors were constrained to equality so that the models were locally identified.

We confirmed three pieces of information prior to running the final analysis. First, we applied univariate FUSION to examine whether the top three g-factor T-SEM hits (*ZSCAN9, PRSS16, ZNF184*) were significantly associated with the disorders (Supplementary Data 7). As PD and ALZ were not significantly associated with any of the hits we did not consider these two traits further. Second, we confirmed that genetic *g* was significantly, genetically correlated with the Internalizing and Psychotic disorders factor, and third that these were not significant $Q_{Gene}$ hits for either psychiatric factor. Finally, we examined the proportion of genetic overlap across *g* and these psychiatric factors that were statistically mediated by gene expression for these top three hits. This was calculated by first estimating separate models for each of the two psychiatric factors and each of the three top hits (6 models in total) in which the gene predicted both *g* and the psychiatric factor, and the residual covariance across these two factors was freely estimated. To obtain a standard error on the estimate, the proportion of genetic overlap mediated by gene expression was calculated as a "ghost" parameter when estimating the model as:

$$\% \ \mathbf{mediated} r_\mathbf{g} = \frac{\mathbf{b}_{\mathbf{Gene,g}} \times \sigma^2_{\mathbf{Gene}} \times \mathbf{b}_{\mathbf{Gene,Psych}}}{\mathbf{b}_{\mathbf{Gene,g}} \times \sigma^2_{\mathbf{Gene}} \times \mathbf{b}_{\mathbf{Gene,Psych}} + \mathbf{r}_{\mathbf{u(g,psych)}}}, \quad (14)$$

where $\mathbf{b}_{\mathbf{Gene,g}}$ is the estimated effect of gene expression on, $\sigma^2_{\mathbf{Gene}}$ is the variance of the gene provided by FUSION, $\mathbf{b}_{\mathbf{Gene,Psych}}$ is the estimated effect of gene expression on the psychiatric factor, and $\mathbf{r}_{\mathbf{u(g,psych)}}$ is the residual genetic covariance across *g* and the psychiatric factor. The denominator of this equation then reflects the total genetic covariance ($r_g$) across *g* and the psychiatric factor.

### Stratified Genomic SEM

Stratified Genomic SEM begins by estimating genetic correlation and covariance matrices stratified across different gene sets and categories (referred to as functional annotations) using a multivariable version of Stratified LDSC[11,21]. The model of interest is then estimated for the

functional annotation including all SNPs. In the context of the current analyses—where the parameters of interest reflect the factor variance of *g*, genetic overlap across *g* and the psychotic disorders factor, and the residual variances of the phenotypes—the factor loadings are subsequently fixed from the estimates obtained from the annotation including all SNPs, and the remaining model parameters are freely estimated within each annotation. The freely estimated parameter and sandwich corrected standard error within a given annotation are then scaled relative to the estimate obtained for the annotation including all SNPs, such that the estimate now reflects a proportion of the total, genome-wide estimate. In the unstandardized case, the enrichment "ratio of ratios" is then calculated by scaling this proportional estimate by the number of SNPs in the corresponding annotation as a proportion of the total SNPs examined. Enrichment is then observed when the proportion of genome-wide variance observed in an annotation is greater than the proportional size of the annotation.

Zero-order, stratified genetic covariance, and correlation matrices were estimated using the s_ldsc function in Genomic SEM. This included 97 functional annotations from the 1000 Genomes Phase 3 BaselineLD Version 2.2 provided by the original S-LDSC authors[21], tissue-specific histone marks from the Roadmap Epigenetics Project[47], and tissue-specific gene expression from GTEx[15] and DEPICT[48]. For tissue-specific gene expression and histone/chromatin marks, we utilized only brain and endocrine relevant regions in addition to 5 randomly selected control regions from each (i.e., 10 controls total). We additionally utilized 29 functional annotations created using data from Genome Aggregation Database (gnomAD)[49] and GTEx[15] to examine the interaction between protein-truncating variant (PTV)-intolerant (PI) genes and human hippocampal and prefrontal brain cells. Details on parameters used to create these 29 annotations can be found in Grotzinger et al. (2022)[11].

Enrichment was not estimated for continuous or flanking window/control annotations, yielding a total of 168 binary annotations. We further remove 13 annotations that were non-positive definite and required smoothing the stratified covariance matrix such that any point estimate in the matrix produced a *Z*-statistic discrepancy >1.96 pre- and post-smoothing. For a Bonferroni correction at <0.05 this corresponds to a significance threshold of $p < 3.22E{-}4$ across the 155 remaining functional annotations. Analyses examining enrichment of the genetic overlap across a *g*-factor with a psychotic disorder factor largely mirrored those for the enrichment of the *g*-factor. When pruning based on the *Z*-statistic discrepancy for smoothing, a total of 16 annotations were removed for this analysis; however, we use the same Bonferroni corrected threshold for 155 tests for comparative purposes.

We additionally estimate enrichment for *g* using stratified genetic correlation (as opposed to covariance) matrices, which can be used to identify annotations for which enrichment of pleiotropic and trait-specific signals are disproportionate. Estimation of enrichment in standardized space (e.g., when using stratified correlation matrices as input) does not require dividing the enrichment estimate by the proportional size of the annotation, as all annotations (including the genome-wide annotation) are on the same scale. We did not observe any significantly enriched annotations for these stratified correlation analyses (Supplementary Data 11; Supplementary Fig. 17). We also did not observe any significant enrichment of the genetic correlation between *g* and the psychotic disorders factors (Supplementary Data 11). We note that these analyses are likely underpowered as significant enrichment in a standardized space requires an annotation that indexes genetic overlap across traits far above and beyond what is observed at the genome-wide level. Therefore, we would characterize our results as reflecting clear genetic risk-sharing within a number of annotations, but inconclusive as to whether these annotations are relevant to both pleiotropic and trait-specific signals.

## Quality control procedures

We refer the reader to the original article describing genetic *g* for details about sample ascertainment, quality control, and related procedures for the seven cognitive tests[9], in addition to the corresponding articles for the analyzed external traits (Supplementary Data 7 for details and references). Default quality control (QC) procedures were used for the *munge* function in Genomic SEM prior to running either LDSC or S-LDSC. This included removing SNPs with an MAF < 1%, information scores (INFO) < 0.9 and filtering SNPs to HapMap3. The LD scores used for the overall LDSC model were estimated from the European sample of 1000 Genomes and excluded the MHC region. LD scores used as input for both LDSC and S-LDSC were also restricted to HapMap3 SNPs as these tend to be well-imputed and produce accurate estimates of heritability. For the analyzed binary traits, the heritability estimates from LDSC were converted to the liability scale prior to running analyses using the same population prevalence employed by the corresponding univariate GWAS for comparative purposes (Supplementary Data 7). We also input a sample prevalence of 0.5 and the sum of effective sample sizes across cohorts contributing to the meta-analytic GWAS for a given trait as we have shown this to produce an unbiased estimate of liability scale heritability in the population[50].

Prior to running FUSION, alleles were aligned across univariate summary statistics to the 1000 Genomes Phase 3 LD reference panel, restricted to SNPS with an MAF > 1%, SNPs with an INFO score > 0.6, and restricted to those SNPs that were present across all seven cognitive tests. Using these QC steps, there were 7,857,346 SNPS present across all tests, of which 1,157,709 SNPs were present in the LD reference panel data and subsequently used by FUSION to produce univariate TWAS estimates. Univariate FUSION summary statistics were subsequently standardized with respect to the total variance in the outcome using the *read_fusion* function in *GenomicSEM*. Standard errors were also corrected for genomic inflation using the conservative approach of multiplying the standard errors by the univariate LDSC intercept when the intercept was above 1.

### Reporting summary

Further information on research design is available in the Nature Research Reporting Summary linked to this article.

## Data availability

The data that support the findings of this study are all publicly available or can be requested for access. Specific download links for various datasets are directly below. Summary statistics for the *g*-factor and the seven, individual cognitive traits are available from: https://datashare.is.ed.ac.uk/handle/10283/3756. Summary statistics for bipolar disorder data can be found here: https://figshare.com/articles/dataset/PGC3_bipolar_disorder_GWAS_summary_statistics/14102594. Summary statistics for schizophrenia can be found here: https://figshare.com/articles/dataset/scz2022/19426775. Summary statistics for the major depressive disorder can be found here: https://datashare.ed.ac.uk/handle/10283/3203. Summary statistics for anxiety can be downloaded here: https://drive.google.com/drive/folders/1fguHvz7l2G45sbMI9h_veQun4aXNTy1v. Summary statistics for Alzheimer's disease can be easily requested here: https://www.niagads.org/datasets/ng00075. Summary statistics for Parkinson's disease can be downloaded here: https://drive.google.com/drive/folders/10bGj6HfAXgl-JslpI9ZJIL_JIgZyktxn. Data from gnomAD used to identify PI genes for the creation of annotations can be downloaded here: https://storage.googleapis.com/gnomad-public/release/2.1.1/constraint/gnomad.v2.1.1.lof_metrics.by_gene.txt.bgz. Gene count data per cell for creation of annotations were obtained from: https://storage.googleapis.com/gtex_additional_datasets/single_cell_data/GTEx_dronc seq_hip_pcf.tar. Data which maps individual cells to cell types (e.g. neuron, astrocyte,

etc.) were obtained from: https://static-content.springer.com/esm/art%3A10.1038%2Fnmeth.4407/MediaObjects/41592_2017_BFnmeth4407_MOESM10_ESM.xlsx. Links to the LD-scores, reference panel data, and the code used to produce the current results can all be found at: https://github.com/GenomicSEM/GenomicSEM/wiki. Links to the BaselineLD v2.2 annotations can be found here: https://data.broadinstitute.org/alkesgroup/LDSCORE/. Links to the reference weights used for FUSION from GTEx and CMC can be found here: http://gusevlab.org/projects/fusion/.

## Code availability

GenomicSEM software (which now includes the T-SEM and Stratified GenomicSEM extensions), is an R package that is available from GitHub at the following URL: https://github.com/GenomicSEM/GenomicSEM. Directions for installing the GenomicSEM R package can be found at: https://github.com/GenomicSEM/GenomicSEM/wiki. The specific code release used for these analyses can be found here: https://zenodo.org/badge/latestdoi/456633204.

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

## Acknowledgements

This work presented here would not have been possible without the enormous efforts put forth by the investigators and participants from Psychiatric Genetics Consortium and UK Biobank. The work from these contributing groups was supported by numerous grants from governmental and charitable bodies as well as philanthropic donations. Research reported in this publication was supported by the National Institute of Mental Health of the National Institutes of Health under Award Number R01MH120219 and the National Institute of Aging under Award Number RF1AG073593. The content is solely the responsibility of the authors and does not necessarily represent the official views of the National Institutes of Health. A.D.G. was additionally supported by NIH Grant R01HD083613. E.M.T.-D. was additionally supported by NIH grants R01AG054628 and R01HD083613 and the Jacobs Foundation. J.F. and E.M.T.-D. are members of the Population Research Center (PRC) and Center on Aging and Population Sciences (CAPS) at The University of Texas at Austin, which are supported by National Institutes of Health (NIH) grants P2CHD042849 and P30AG066614, respectively. M.G.N. is additionally supported by ZonMW grants 849200011 and 531003014 from The Netherlands Organization for Health Research and Development, a VENI grant awarded by NWO (VI.Veni.191 G.030) and is a Jacobs Foundation Fellow.

## Author contributions

Study design: A.D.G., J.F., G.D., M.G.N., E.M.T.-D. Methods development: A.D.G., M.G.N., E.M.T.-D. Software development: A.D.G., M.G.N. Simulation studies: A.D.G., E.M.T.-D. Writing, feedback, and editing: A.D.G., J.F, G.D., M.G.N., E.M.T.-D.

## Competing interests

The authors declare no competing interests.
