## [Peer Review File · Nature Communications]

Transcriptome-wide and Stratified Genomic Structural Equation Modeling Identify Neurobiological Pathways Shared across Diverse Cognitive TraitsReviewers' comments:

Reviewer #1 (Remarks to the Author):

This manuscript introduces a new method, Transcriptome-wide SEM or T-SEM, to perform a transcriptome-wide scan for gene associations with a common factor (g) underlying genetically correlated traits. This method is built upon their previously published high-impact method, Genomic SEM, with application on tissue-specific transcription-wide association studies (TWAS) for cognitive traits. They first show by simulation that T-SEM is well-powered to identify genes that affect the g-factor underlying multiple cognitive traits as well as genes that differentially affect individual traits detected by Q_Gene Statistic. They apply the method to 7 cognitive traits in UK Biobank data and find a number of genes whose tissue-specific expression is associated with the g-factor, including 10 novel genes that are not detected by standard univariate TWAS. A follow-up stratified Genomic SEM analysis identifies functional annotations that are enriched in genes sharing across traits. Overall, the study is sound, and their method should be of interest to a broad range of researchers in complex trait genetics and functional genomics. I hope the authors find the following comments useful.

1. Just like Genomic SEM is innovative in detecting genotype associations with the unobserved common factor across traits, the main novelty of T-SEM appears to be the capacity of detecting such associations at gene level, by first running TWAS for each trait and taking the result as input for T-SEM. However, since they use the same eQTL datasets for all traits in the TWAS analysis, it seems an equivalent (at least similar) approach is to first run Genomic SEM (or a multivariate GWAS) on the 7 cognitive traits to generate SNP association statistics and then perform summary-data-based TWAS with tissue-specific eQTL weights. Can the authors comment on the advantages/differences of the T-SEM over this approach?
2. The evaluate the performance of T-SEM with different simulation scenarios. However, the simulation is only based on the observed result of the top-associated gene in the real data analysis. They observe zero false positive in different scenarios, but this could be just because the power to identify the correct model is very high for this particular gene. I wonder if the same result would hold for a random gene.
3. It might be worth to show the pairwise genetic correlations between the 7 cognitive traits. This may help to understand the gene discovery results. For example, they identify similar total number of genes associated with common g (76) or heterogeneously across traits (62). Does it also reflect on the level of genetic correlations between the traits? Some Q_Gene statistics are found to be driven mainly by Reaction time. Is this cognitive trait less genetically correlated with others?
4. They find 76 unique genes that have significant effects on the common g. I wonder if it is possible to estimate how much variance in g is explained by the predicted expression of these identified genes.
5. From Figure 1, it looks like there are some loci with both significant g-factor associations and Q_gene signals, for example, the big peaks on chromosomes 3 and 17, and other smaller but nearly symmetric peaks on other chromosomes as well. Are the genes with consistent and inconsistent association patterns with g overlapped or in proximity? If so, what's the explanation for this?
6. In Discussion, the authors predict that their top multivariate TWAS hit, ZSCAN9, may be associated with neurodegenerative diseases, such as Alzheimer's disease (AD), and another top hit, ZNF184, is associated with Parkinson's disease progression and depression. It seems these hypotheses can be also tested by including these diseases in their framework and see if this gene is associated with the common g underlying both cognitive traits and diseases.
7. They do a gene-set analysis based on the detected genes with both the g-factor and Q_Gene associations to identify pathways, and also perform pathway enrichment by stratified Genomic SEM. It

is not quite clear how consistent of the two sets of results based on T-SEM and Genomic SEM analysis.

8. Is it possible to write/explain their model in the form of linear mixed model? I think it would be helpful for readers to better understand their method if they can link the factor analysis to linear models as many researchers in the field are more familiar with the latter.

Reviewer #2 (Remarks to the Author):

This paper is to develop and validate Transcriptome-wide Structural Equation Modeling (T-SEM), a novel method for studying the effects of tissue-specific gene expression within a multivariate space. Although it has several interesting findings, this reviewer has several major concerns about this paper.

1. The T-SEM can be regarded as a straightforward extension of Genomic SEM (Grotzinger et al., 2019, Nature Human Behavior). The only difference is to replace SNP with Gene in the GSEM. The estimation procedure is the standard two-stage WLS procedure. Therefore, from statistical perspective, the contributions are at most marginal.

2. Most references have not been cited appropriately throughout the paper. To name a few, for instance, the reference for GSEM should be [20], whereas the authors cited [21] on page 9. Please see "T-SEM estimation follows the general two-stage approach introduced in the Genomic SEM framework,21". The reference [21] is not about [20]. This reviewer checked all major references, including the reference for FUSION, and found that most references are incorrectly cited.

3. Although the authors gave some details about T-SEM, many key steps strongly depend other methods so that this reviewer has to read more than 10+ papers to understand the proposed methods. Moreover, stratified Genomic SEM does not have details so that readers could hardly understand.

4. The whole paper is not well written. The logic does not flow very well. For instance, the second paragraph does not connect with the first and third paragraphs very well.

RESPONSE TO REVIEWERS

Reviewer #1:

This manuscript introduces a new method, Transcriptome-wide SEM or T-SEM, to perform a transcriptome-wide scan for gene associations with a common factor (g) underlying genetically correlated traits. This method is built upon their previously published high-impact method, Genomic SEM, with application on tissue-specific transcription-wide association studies (TWAS) for cognitive traits. They first show by simulation that T-SEM is well-powered to identify genes that affect the g-factor underlying multiple cognitive traits as well as genes that differentially affect individual traits detected by Q_{Gene} Statistic. They apply the method to 7 cognitive traits in UK Biobank data and find a number of genes whose tissue-specific expression is associated with the g-factor, including 10 novel genes that are not detected by standard univariate TWAS. A follow-up stratified Genomic SEM analysis identifies functional annotations that are enriched in genes sharing across traits. Overall, the study is sound, and their method should be of interest to a broad range of researchers in complex trait genetics and functional genomics. I hope the authors find the following comments useful.

1. Just like Genomic SEM is innovative in detecting genotype associations with the unobserved common factor across traits, the main novelty of T-SEM appears to be the capacity of detecting such associations at gene level, by first running TWAS for each trait and taking the result as input for T-SEM. However, since they use the same eQTL datasets for all traits in the TWAS analysis, it seems an equivalent (at least similar) approach is to first run Genomic SEM (or a multivariate GWAS) on the 7 cognitive traits to generate SNP association statistics and then perform summary-data-based TWAS with tissue-specific eQTL weights. Can the authors comment on the advantages/differences of the T-SEM over this approach?

We thank the reviewer for pointing this out as many readers would likely have the same response. We now make explicit both in our writing and via simulation that a critical advantage of T-SEM over performing an ad hoc, univariate TWAS of the GWAS summary statistics from a multivariate analysis is the ability to estimate Q_{Gene} to determine whether the effects of gene expression on the constellation of phenotypes plausibly acts via the general factor(s). When the ad hoc approach is implemented, gene expression may be implicated as relevant for a general factor in circumstances in which it is only relevant to for a subset of GWAS phenotypes composing that factor. In contrast, T-SEM directly tests whether the factor model appropriately represents the pattern of effects of the tissue-specific gene expression on the factor. When Q_{Gene} is significant, this indicates that the tissue-specific gene expression acts unevenly on the traits, potentially via trait-specific pathways. To illustrate and validate this point, we have added an extensive set of simulations that compare these two approaches by generating data at the level of the individual SNPs that comprise the eQTL weights for a given gene. Our simulation includes conditions under which the population TWAS effects for the individual GWAS phenotypes perfectly match a model of gene expression operating through the common factor, and conditions under which the individual TWAS effects diverge to varying degrees from the expectations of the factor model. We find that TWAS and T-SEM both produce concordant results, but that T-SEM uniquely guards against false positives and identifies sources of differentiation across genetically correlated traits via Q_{Gene} . We now write in the main text (p. 5):

“We went on to perform a second set of simulations designed to compare the performance of TWAS of summary statistics from multivariate GWAS relative to results obtained using T-SEM of FUSION output. These simulations began by specifying population effects at the SNP-level that were weighted by the functional weights from FUSION (see **Method** for details). Simulations were again conducted for scenarios that varied in the degree to which the population generating TWAS effects were reflective of a model in which gene expression operates entirely via the common factor. When the population generating effects were consistent with the expectations implied by the common factor model, we observe 100% power at a Bonferroni corrected threshold for both TWAS of the common factor summary statistics and T-SEM of FUSION output (Supplementary Table 3). Q_{Gene} was also well-calibrated in this scenario with a 5% FPR at $\alpha=.05$. When the population SNP effects were set to 0, we similarly observe a well-controlled FPR, with 4% and 3% of runs significant at $p < .05$ for TWAS and T-SEM, respectively, and 7% significant for Q_{Gene} . Finally, we find that when the population generating TWAS effects strongly deviate from the factor model, 100% of simulations for both TWAS and T-SEM were significant at a Bonferroni corrected threshold. However, Q_{Gene} estimated within the T-SEM framework also showed 100% power, thereby safeguarding against false positive inference that the effects of gene expression operate at the level of the factor underlying the individual GWAS traits, and appropriately identifying gene expression patterns responsible for trait differentiation. Indeed, TWAS of the common factor summary statistics and T-SEM of FUSION output displayed strong concordance in estimates for the different population generating scenarios (Supplementary Figure 6 for scatter plots; Supplementary Figure 7 for QQ-plots), but only T-SEM was able to produce the Q_{Gene} statistic necessary to safeguard against false positive inference and identify trait-specific pathways of TWAS effects.”

We go on to provide greater detail in the Online Method (ps. 16-17):

“SNP-level Simulations

Simulation Procedure. Our next of simulations sought to compare the performance of TWAS of common factor, GWAS summary statistics generated using Genomic SEM relative to results obtained using T-SEM of FUSION output for the GWAS phenotypes that define the common factor. These simulations began by generating 100 sets of genetic covariance and sampling covariance matrices following the procedure described in de la Fuente et al. (2021).⁵⁴ Data were simulated using European population LD scores for 1,184,461 HapMap3 SNPs, excluding the MHC region, for five continuous traits. These traits were specified to be 20% heritable in the population, genetically and phenotypically correlate at .7, and have minimal population stratification (univariate LDSC intercept = 1.04). All traits had a sample size of 150,000, with no sample overlap across the traits. These GWAS summary statistics were simulated from the multivariable LDSC equation as:

$$[Z_{1j}, Z_{2j}, \dots, Z_{5j}] \sim N([0, 0, \dots, 0], \text{cov}(Z_{1j}, Z_{2j}, \dots, Z_{5j}))$$

where,

$$\text{cov}(Z_{1j}, Z_{2j}, \dots, Z_{5j}) = \begin{bmatrix} N_1 \frac{h_1^2}{M} \ell(j) + 1 + a_1 & \square & \square & \square \\ \sqrt{N_1 N_2} \frac{\sigma_{g1,2}}{M} \ell(j) + \frac{\rho_{1,2} N_{s1,2}}{\sqrt{N_1 N_2}} & N_2 \frac{h_2^2}{M} \ell(j) + 1 + a_2 & \square & \square \\ \vdots & \vdots & \ddots & \square \\ \sqrt{N_1 N_5} \frac{\sigma_{g1,5}}{M} \ell(j) + \frac{\rho_{1,10} N_{s1,5}}{\sqrt{N_1 N_5}} & \sqrt{N_2 N_5} \frac{\sigma_{g2,5}}{M} \ell(j) + \frac{\rho_{2,5} N_{s2,5}}{\sqrt{N_2 N_5}} & \dots & N_5 \frac{h_5^2}{M} \ell(j) + 1 + a_5 \end{bmatrix}$$

and $[Z_{1j}, Z_{2j}, \dots, Z_{5j}]$ reflects the Z statistics for the five GWAS cohorts, N is the sample size of the individual GWAS that was set to 150,000 for all five phenotypes, N_s is the number of overlapping individuals (0 in this case), M is the number of SNPs from the LD file (1,184,461), ρ is the phenotypic correlation (set to 0.7 here, but inconsequential when there is no sample overlap), $\ell(j)$ is the LD score of SNP j , and $a+1$ reflects the univariate LDSC intercept indexing unmeasured confounds such as population stratification (1.04). The bivariate LDSC intercept, expressed as $\frac{\rho_{1,2} N_{s1,2}}{\sqrt{N_1 N_2}}$ for phenotypes 1 and 2, reduced to 0 for all pairs of phenotypes as the sample overlap was 0. These simulated GWAS summary statistics were then used as input to multivariable LDSC to produce 100 sets of genetic covariance matrices and associated sampling covariance matrices, and subsequently paired with simulated SNP effects for the GNL3 gene for five population generating scenarios.

The population-level SNP effects on the five phenotypes (b_{GWAS}) were computed using three, key population parameters: (i) the SNP effect on gene expression ($b_{e\text{QTL}}$), (ii) the effect of gene expression on the common factor (γ), and (iii) the effect of the common factor on the phenotypes (i.e., the factor loadings; λ_i). The population SNP effects on gene expression were taken from the pre-compiled weights provided by FUSION. We specifically use the top1 weights as this reflects the best performing model for GNL3 in the basal ganglia. The top1 weights refer to the single best eQTL weights, where only the individual SNP with the largest weight is used for TWAS, in this case rs1108842. The top1 weight is computed by FUSION as:

$$\text{top1} = \frac{G\mathcal{E}}{\sqrt{n-1}},$$

where G is a matrix of standardized genotypes, \mathcal{E} is a vector of standardized gene expression from the same participant sample, and n is the number of samples used to calculate the weights, which was 144 for GNL3 expression in the basal ganglia. For the purposes of the current simulations, the top1 weight was converted to a partially standardized, population $b_{e\text{QTL}}$ (i.e., standardized with respect to gene expression, but not standardized with respect to the SNP variance). This was calculated as:

$$b_{e\text{QTL}} = \frac{\text{top1}}{\sigma_{\text{SNP}} \sqrt{n-1}},$$

where σ_{SNP} is the standard deviation of the SNP calculated from the 1000 Genomes Phase 3 European reference panel as .706 for rs1108842. These calculations then yielded a population $b_{e\text{QTL}}$ of .442. The true effect (γ) of tissue-specific gene expression on the common factor was set to reflect the beta coefficient (scaled from a model using unit variance identification) from the real data T-SEM results of GNL3 for the g-factor ($b = -.089$), as this reflects a realistic point estimate. The unstandardized population effect of the common factor on the phenotypes (λ_i , also scaled using unit variance identification) was .374 using the population parameters described above. Putting these pieces together, $b_{\text{GWAS},i}$ for each trait, i , in the population was calculated as:

$$b_{\text{GWAS},i} = b_{e\text{QTL}} * \gamma * \lambda_i.$$

Observed SNP effects for each phenotype were then sampled from the sampling distribution given as:

$$[b_{GWAS1}, b_{GWAS2}, \dots, b_{GWAS5}] \sim N$$

$$\left([b_{GWAS1}, b_{GWAS2}, \dots, b_{GWAS5}], \begin{bmatrix} \frac{1-2MAF(1-MAF)b_{GWAS1}^2}{2MAF(1-MAF)n_1} & & & & \\ 0 & \frac{1-2MAF(1-MAF)b_{GWAS2}^2}{2MAF(1-MAF)n_2} & & & \\ \vdots & \vdots & \ddots & & \\ 0 & 0 & \dots & \frac{1-2MAF(1-MAF)b_{GWAS5}^2}{2MAF(1-MAF)n_5} & \end{bmatrix} \right)$$

Note that the off-diagonals of the sampling covariance matrix for the betas was 0, as expected under conditions of 0 sample overlap (otherwise, the off diagonal elements are determined by the sampling correlation, equal to the cross trait LDSC intercept, as given earlier, rescaled to covariances using the corresponding sampling variances that are on the diagonals). The population betas, and the corresponding sampling distribution, were perturbed from their expectations above to reflect five population generating scenarios. Scenario 1 reflected one in which gene expression operated entirely through the common factor, such that the population betas were calculated as described (i.e., no perturbation of population betas). Scenario 2 deviated from the factor model where the direction of the population betas was reversed for three of the five phenotypes, and Scenario 3 deviated still further where the direction was reversed, and the population effect doubled for three of the five phenotypes. For Scenario 4 the population betas were set to 0 for three of the five phenotypes, and for Scenario 5 the population betas were set to 0 for all phenotypes. We sampled 100 sets of SNP effects for each of the five population generating scenarios.

These simulated SNP-phenotype betas and their standard errors were then used in analytic pipelines both for T-SEM and for an ad hoc univariate TWAS of summary statistics from multivariate GWAS. The T-SEM pipeline involved inputting the simulated SNP-phenotype Z-statistics for each phenotype to FUSION, converting these phenotype-level FUSION estimates to gene-phenotype covariances, and appending these FUSION estimates to the simulated genetic covariance and sampling covariance matrices. These matrices were then used as input to T-SEM to produce both estimates of gene expression on the common factor and the Q_{Gene} heterogeneity estimate. To produce ad hoc TWAS estimates for the common factor, the SNP-phenotype betas were converted to SNP-phenotype covariances so that they could be appended to the 100 simulated genetic covariance and sampling covariance matrices. These expanded matrices were then used as input into Genomic SEM to produce estimates of SNP effects on the common factor, and the common factor GWAS summary statistics were then used as input for univariate analysis in FUSION.

Simulation Results. Results for Scenario 1, for which the population generating parameters perfectly matched a model of tissue-specific gene expression operating through the common factor, revealed 100% power for both TWAS and T-SEM using a Bonferroni corrected threshold of $p < 9.46E-7$ and, as would be expected, 0% positivity for Q_{Gene} (Supplementary Table 3). For Scenario 2, characterized by the population direction SNP effect reversed for three of the five phenotypes, there was 0% positivity for TWAS and T-SEM and 100% power for Q_{Gene} . Scenario 3 then deviated still further from the factor model with the direction of SNP effects reversed and doubled for three of the five phenotypes. In this instance, the heterogenous population effects were large enough that we observe 100% positivity for TWAS and 97% positivity for T-SEM. Critically, we find also that Q_{Gene} has 100% power. This scenario demonstrates

the key advantage of the T-SEM framework relative to performing a TWAS of multivariate GWAS summary statistics produced from multivariate methods like Genomic SEM. That is, while we observe strong concordance between TWAS and T-SEM estimates (Supplementary Table 3; Supplementary Figure 6), only T-SEM can employ Q_{Gene} as a means of both guarding against false positives and identifying patterns of gene expression that are specific to a trait or subset of traits. For Scenario 4 and 5 that simulated population betas of 0 for three and five of the phenotypes, respectively, the signal was highly attenuated and did not evince observable deviations from the null (Supplementary Figure 7 for QQ-plots). We observe 7% positivity for TWAS and T-SEM for Scenario 4, with 2 of the 7 T-SEM hits identified as significant for Q_{Gene} , and 9% power for Q_{Gene} . Finally, for Scenario 5 we observe 0% positivity across TWAS, T-SEM, and Q_{Gene} .

We end by considering the false positive rate (FPR) at $p < .05$ for the two scenarios for which the population effects can be clearly considered as reflective of a given null hypothesis. More specifically, Scenario 1 that perfectly matches the common factor model reflects the null for Q_{Gene} , and in this case we observe a 5% FPR. In addition, Scenario 5 is reflective of the null for both the common factor signal and Q_{Gene} for our specific simulating parameters where the equal factor loadings across the phenotypes mirror a pattern of population betas of 0s across the phenotypes. Once again, we find a well-controlled FPR, with 4% FPR for TWAS, 3% for T-SEM, and 7% for Q_{Gene} . In summary, SNP-level simulation results indicate that positivity rate (i.e. power) for TWAS and T-SEM appropriately scales as a function of the size of population effects and the degree to which the effects of gene expression on the individual traits correspond to the expectations of the factor model, that FPR is well controlled for population scenarios that reflect a given null, and that TWAS and T-SEM yield a concordant set of association results, with the critical exception that the Q_{Gene} statistic estimated within T-SEM guards against false positives and identifies sources of differentiation of gene expression effects across traits.”

2. The evaluate the performance of T-SEM with different simulation scenarios. However, the simulation is only based on the observed result of the top-associated gene in the real data analysis. They observe zero false positive in different scenarios, but this could be just because the power to identify the correct model is very high for this particular gene. I wonder if the same result would hold for a random gene.

We purposely chose to use a top associated gene so as to increase the potential for inflation in the the false positive rate for those simulating conditions where the gene affects only a subset of the g-factor traits. Our key illustrative point is that, even under conditions in which false positives are prevalent, we are able to safeguard against them using the Q_{Gene} within T-SEM (whereas they are uncontrolled under the ad hoc approach described earlier).

With that said, we completely agree that these simulations are informative with respect to the specific population generating conditions, and that it provides greater context for understanding the performance of T-SEM to use a far less significant gene. To this end, we have added another set of simulations for the gene in the 50th percentile of g-factor, T-SEM p-values (ZNF749 in the Hippocampus with a real data p-value of .362). These results show a similar pattern of relationships across the simulating scenarios, but as would be expected also evince a largely null signal consistent with population generating values for a gene with null signal. For the simulations at the gene expression level, we have updated the main text to highlight both the previously reported results for a top hit and this 50th percentile gene. We now write on ps. 4-5 of the main text:

“Validation of T-SEM via Simulation

We began by running two sets of simulations to validate the calibration of T-SEM. The first set of simulations generated patterns of gene expression across seven, population-generating conditions. For each condition, separate datasets were simulated for both a top hit and for a gene from the 50th percentile of the p-value distribution in our empirical analysis of the g-factor. All results reported below follow the same pattern across conditions for both sets of population gene expression effects, apart from the expected decrease in signal for the 50th percentile gene relative to the top hit (**Method**; Supplementary Figures 2-5; Supplementary Table 2). The population-generating parameters for the first condition reflected those implied by a model of gene expression operating entirely through a common factor model of genomic g. This then reflects a scenario in which the power to detect gene effects on the factor is expected to be high and the signal for Q_{Gene} to conform to a null distribution. Indeed, this is what we observe (Supplementary Table 2), indicating power for discovery and appropriate Type I Error control. The remaining conditions were specified such that the pattern of effects for a gene on the given traits in the population increasingly deviated from the expectations of the common factor model. Results confirmed that as the simulated effect shifts away from the expectation under a common factor model, the power to detect gene expression effects on a common factor and for Q_{Gene} decreased and increased, respectively. Simulation results further revealed that T-SEM associations were not merely a recapitulation of the associations for the most well-powered univariate trait that loads on the factor, that there is a well-controlled false positive rate (FPR) of < 5% at $p < .05$ when the gene expression effects on the traits is 0 in the population, and that power to detect effects for Q_{Gene} is greatest when the gene effects on the individual traits reflect a mixture of heterogeneous associations that deviate from the expectations of the factor model.”

In addition, we have updated the Online Method to report the simulation procedure and a more detailed outline of the results for both the prior results for a top gene and the 50th percentile results. We **highlight in bold** for the reviewer responses only the added sections that are of particular relevance to the 50th percentile results, and specifically write on ps. 13-14:

“Simulations of Gene Expression

Simulation Procedure. As a first step in the simulation procedure, a model implied genetic covariance matrix was calculated for a model in which gene expression from real data analyses was specified to predict the g-factor. This reflects a best-case scenario for common factor signal in T-SEM as the model implied matrix reflects a pattern of relationships that operate entirely through the common factor. **We utilized two model implied matrices, one from gene expression results from the top hit (ZSCAN9 expression in the cerebellum), and the other from results for the gene with null signal reflecting the 50th percentile real data results for the g-factor, T-SEM p-values (ZNF749 in the hippocampus).** Seven versions of these two model implied matrices were subsequently used to construct population generating covariance matrices, from which 1,000 genetic covariance matrices were then sampled using *rmvnorm* in R for each of the seven scenarios (i.e., 14,000 total simulated datasets). These seven scenarios consisted of: Scenario 1, reflective of the alluded to best-case scenario where the model implied matrix was unchanged; Scenario 2 in which the covariance between the gene and RT (the phenotype with the smallest factor loading) was set to 0; Scenario 3 in which the covariance between the gene and Trails-b (the phenotype with the largest factor loading) was set to 0; Scenario 4 in which the covariance between the gene and all phenotypes except trails-B was set to 0; Scenario 5 in which the covariance between the gene and all phenotypes except RT was set to 0; Scenario 6 in which the covariance between the gene and all phenotypes was set to 0; and Scenario 7 in which the directionality of the covariance between the gene and matrices, memory, and RT was reversed.

The observed sampling covariance matrix, $V_{S_{Full}}$, was used both to sample from the population matrices for each condition and was paired with each simulated genetic covariance matrix for the real data analyses. This has the advantage of providing a set of simulations that are both directly relevant to interpretation of the current analyses and, by definition, reflective of the types of real-data scenarios where T-SEM might be applied. All simulations applied the same Bonferroni corrected threshold for significance as used in the real-data analysis.

Simulation Results. These scenarios were selected to reflect a gradient of deviations from the factor model, where Scenario 1 exactly matches the common factor model and Scenario 7 reflects a strong departure from the model wherein gene expression has directionally opposing effects across the g-factor phenotypes. This allowed us to test whether T-SEM appropriately down weights estimates of gene expression effects on the common factor in a graded fashion as the generating population gradually shifts further away from the common factor structure. Results for Scenario 1, with population gene effects specified to operate solely through the common factor, revealed hits on the common factor for all but 1 run for the ZSCAN9 (top hit from real data) simulations (i.e., 99.6% hits; Supplementary Table 2) and evinced the strongest signal across scenarios (Supplementary Figure 4). The signal was slightly reduced for Scenario 2, in which the gene effect was set to 0 for the phenotype with the smallest loading (RT) and more attenuated for Scenario 3, in which the gene effect was set to 0 for the phenotype with the largest loading (Trails-b; Supplementary Figure 2). It is also worth noting that Scenario 3 still produced hits for 82.7% of the top hit simulations. As expected, the signal was greatly reduced relative to these first three scenarios for Scenario 4, in which the gene effect was set to 0 for all but Trails-b, and further reduced for Scenario 5, in which the gene effect was set to 0 for all but RT, with no simulations producing hits for either scenario. Taken together, this demonstrates that phenotypes with larger factor loadings are appropriately weighted when estimating effects of gene expression and that the signal for the common factor is by no means a recapitulation of the signal for the strongest phenotype. For Scenario 6, in which the gene expression effect in the population was 0 across all phenotypes, simulation results revealed a null signal, with a well-controlled FPR at $p < .05$ of 4.96% for the top hit simulations and 3.78% for the null signal simulations. Finally, for Scenario 7, in which the directionality of the gene effect was reversed for three of the phenotypes, 35.1% of the runs were identified as significant for the common factor for the top hit simulations. Importantly, all significant runs in this scenario were also identified as hits for Q_{Gene} , which we consider next.

The pattern of results of Q_{Gene} were also consistent with expectation. For Scenario 1, there was null signal, with no simulations identified as Q_{Gene} hits, and an FPR at $p < .05$ of 3.90% for top hit simulations and 6.70% for the null signal simulations. These Scenario 1 results are consistent with the fact that the population did not deviate from the factor structure (Supplementary Figures 3 and 5). The next closest signal was for Scenario 6 in which all associations were set to 0 in the population, reflecting the fact that a pattern of null associations across the phenotypes largely aligns with a common factor model for highly correlated traits. Scenarios 2 and 5, in which the population effect of the gene on RT and all phenotypes except RT was set to 0, respectively, evinced a similar Q_{Gene} signal that was slightly stronger than Scenario 6, but did not produce any Q_{Gene} hits. These results reflect the relatively smaller influence of RT (the phenotype with the smallest loading) on the overall factor model. Scenario 3, in which the population effect of the gene on Trails-B was 0, evinced the next highest signal, with 39.4% of runs producing Q_{Gene} hits for the top hit simulations, followed by Scenario 4, in which the gene effect on all phenotypes except Trails-B was set to 0 in the population, with 46.1% of top hit runs producing Q_{Gene} hits. As expected, Scenario 7, which deviated the strongest from the model with directionally opposing gene effects on matrices, memory, and RT relative to the remaining phenotypes, showed the strongest Q_{Gene} signal by far, with 99.6% of top hit simulations identifying a Q_{Gene} hit. **As would be expected, none**

of the simulations produced factor or Q_{Gene} hits at a Bonferroni corrected threshold for the generating population that reflected the 50th percentile of the real data, g-factor results. However, the patterning of results across conditions mirrored that for the top hit simulation results.”

3. It might be worth to show the pairwise genetic correlations between the 7 cognitive traits. This may help to understand the gene discovery results. For example, they identify similar total number of genes associated with common g (76) or heterogeneously across traits (62). Does it also reflect on the level of genetic correlations between the traits? Some Q_{Gene} statistics are found to be driven mainly by Reaction time. Is this cognitive trait less genetically correlated with others?

We agree that this is a useful metric to consider when interpreting results and have added the genetic heatmap with pairwise point estimates to Figure 1. The reviewer is correct to point out that the results for reaction time (RT) can be understood as reflecting a more unique genetic signal relative to the other traits, which is evidenced by the lower genetic correlation across RT and the other six traits, along with RT's correspondingly low factor loading in the g-factor model.

4. They find 76 unique genes that have significant effects on the common g. I wonder if it is possible to estimate how much variance in g is explained by the predicted expression of these identified genes.

We agree with the reviewer that variance explained in genetic g is a very informative metric to include. To this end, we have updated Supplementary Table 4 to report the genetic variance in g accounted for by each gene expression hit, and now write in the main text (p. 6):

“T-SEM analyses revealed 218 hits for tissue-specific gene expression associated with g at a Bonferroni corrected threshold that explained, on average, 0.13% (range = 0.11% - 0.25%) of the total genetic variance in g (Figure 1; Table 1; Supplementary Table 4).”

*We note also that the conditional and colocalization analyses presented in the paper provide some insight into this question. In particular, the conditional analyses conducted within FUSION speak to the variance explained in nearby GWAS estimates for g by significant genes, but are scaled to reflect only the variance explained within a particular locus window, and therefore do not provide a direct answer to the question of **total** variance explained in genetic g by gene expression across all genes in all tissues.*

g-factor associations and Q_{gene} signals, for example, the big peaks on chromosomes 3 and 17, and other smaller but nearly symmetric peaks on other chromosomes as well. Are the genes with consistent and inconsistent association patterns with g overlapped or in proximity? If so, what's the explanation for this?

This is a very good question. Indeed, some of the Q_{Gene} hits are directly adjacent to g hits, in particular on chromosome 3 as the reviewer highlights. For example, the base pair range for MST1 (a hit for g but not Q_{Gene}) is 49721397 to 49726931, while the base pair range for RNF123 (a Q_{Gene} hit) starts only four base pairs down with a range of 49726935 to 49753910. One way of understanding how to interpret these results is to extend the conditional analyses to include the g-factor estimates for genes significant for both g and Q_{Gene} as a tentative means of quantifying whether physically proximal g and Q_{Gene} hits can be considered as independent signals. To this end, we have updated Supplementary Table 5 to include the conditional results when incorporating those Q_{Gene} hits and now write in the Results section (p. 6):

“As can be visually observed in the Miami plot in Figure 1, there were sets of g hits that were physically proximal to genes significant for both g and Q_{Gene} (e.g., on chromosome 3). With this in mind, we went on to rerun joint analyses utilizing the full set of 218 hits, including the 34 additional genes significant for g and Q_{Gene} . We find that 20 of the 29 genes that were significant from the 184 hit joint analyses remained significant (Supplementary Table 5).”

As nearby genes may code for distinct biological processes, but still be highly correlated such that power is reduced in these cases, we treat these results as tentative. As we write in the Online Method (p. 18):

“It is of note that genes that are estimated to be jointly significant are not necessarily causal and those that are not jointly significant may still be causal. This is due to the fact that the gene expression features in the latter case may simply be characterized by high correlations with other features.”

6. In Discussion, the authors predict that their top multivariate TWAS hit, ZSCAN9, may be associated with neurodegenerative diseases, such as Alzheimer’s disease (AD), and another top hit, ZNF184, is associated with Parkinson’s disease progression and depression. It seems these hypotheses can be also tested by including these diseases in their framework and see if this gene is associated with the common g underlying both cognitive traits and diseases.

We found this to be a very useful suggestion and have added a new set of analyses that examine the proportion of genetic overlap across g and clinically relevant correlates that is mediated by shared patterns of tissue-specific gene expression. These analyses are first reported in the main text on ps. 6-7:

*“We went on to examine whether the three top hits for g (ZSCAN9, PRSS16, ZNF184) explained a significant proportion of the genetic overlap across g and its clinically relevant correlates. We focus here on findings for an Internalizing disorders factor defined by GWAS summary statistics from major depressive disorder²⁷ and anxiety disorders²⁸ and a Psychotic disorders factor defined by bipolar disorder²⁹ and schizophrenia³⁰ (see **Method** for details and results for additional traits). We confirm first that genetic g is significantly, genetically correlated with both the Internalizing disorders factor ($r_g = -.17$, $SE = .03$, $p = 1.33E-8$) and Psychotic disorders factor ($r_g = -.40$, $SE = .03$, $p = 6.45E-51$), that the three top gene expression hits are significantly associated with the individual traits defining these psychiatric factors (Supplementary Table 7), and that these are not Q_{Gene} hits for the psychiatric factors (Supplementary Table 8). Finally, we find that all three hits explained a significant proportion of the genetic overlap across g and these factors (Supplementary Table 8), with the largest effect observed for ZSCAN9 for both the Internalizing (% mediated $r_g = 1.39\%$, $SE = .21$, $p = 3.25E-11$) and Psychotic disorders factor (% mediated $r_g = 1.14\%$, $SE = .14$, $p = 7.39E-16$).”*

We also write in the Discussion (p. 9):

“Major depressive disorder,⁴⁷ anxiety disorders,⁴⁸ schizophrenia,⁴⁹ and bipolar disorder⁵⁰ have all been associated with lower cognitive function. In addition, major depressive disorder⁴¹ and schizophrenia⁵¹ have been associated with the two of the top g hits from the current analyses: ZNF184 and PRSS16, respectively. As a natural extension of these convergent findings, we find that top gene expression hits on g explained a significant proportion of the genetic overlap across g and these psychiatric traits. Building on patterns of enrichment in neuronal subtypes identified for g that mirrored recently described enrichment across bipolar disorder and schizophrenia,²¹ we find also that excitatory prefrontal cortex neurons were enriched in their contribution to the genetic covariance between g and these two disorders. Collectively, these may represent specific biological pathways underlying the well-established association between cognitive impairment and risk for different clusters of psychiatric conditions.”

Finally, we provide additional details in the Online Method (ps. 19-20):

“Gene Expression Mediation of Overlap with Clinically Relevant Correlates

Both g and the top gene expression hits for g have been associated with several clinically relevant outcomes. This includes previously described associations for Alzheimer’s disease and ZSCAN9,³⁹ for major depressive disorder (MDD)⁴¹ and Parkinson’s disease (PD)⁴² and ZNF184, and for schizophrenia (SCZ)⁵¹ and PRSS16. Following up on this work, we examined whether these patterns of gene expression explained a significant proportion of the genetic overlap across g and these correlates of g by utilizing publicly available GWAS summary statistics for ALZ,⁵⁸ PD,⁵⁹ MDD²⁷ and SCZ.³⁰ In line with the multivariate framework employed here, we examine MDD and SCZ in conjunction with GWAS summary statistics for anxiety disorders (ANX)²⁸ and bipolar disorder (BIP),²⁹ respectively, given high levels of genetic overlap across these pairs of disorders. This was done by specifying a two-phenotype Internalizing disorders factor consisting of MDD and ANX and a Psychotic disorders factor consisting of BIP and SCZ; we note that the loadings on these two-phenotype factors were constrained to equality so that the models were locally identified.

We confirmed three pieces of information prior to running the final analysis. First, we applied univariate FUSION to examine whether the top three g -factor T-SEM hits (ZSCAN9, PRSS16, ZNF184) were significantly associated with the disorders (Supplementary Table 7). As PD and ALZ were not significantly associated with any of the hits we did not consider these two traits further. Second, we confirmed that genetic g was significantly, genetically correlated with the Internalizing and Psychotic disorders factor, and third that these were not significant Q_{Gene} hits for either psychiatric factor. Finally, we examined the proportion of genetic overlap across g and these psychiatric factors that was statistically mediated by gene expression for these top three hits. This was calculated by first estimating separate models for each of the two psychiatric factors and each of the three top hits (6 models in total) in which the gene predicted both g and the psychiatric factor, and the residual covariance across these two factors was freely estimated. In order to obtain a standard error on the estimate, the proportion of genetic overlap mediated by gene expression was calculated as a “ghost” parameter when estimating the model as:

$$\% \text{ mediated } r_g = \frac{b_{Gene,g} \times \sigma_{Gene}^2 \times b_{Gene,Psych}}{b_{Gene,g} \times \sigma_{Gene}^2 \times b_{Gene,Psych} + r_{u(g,psych)}}$$

where $b_{Gene,g}$ is the estimated effect of gene expression on g , σ_{Gene}^2 is the variance of the gene provided by FUSION, $b_{Gene,Psych}$ is the estimated effect of gene expression on the psychiatric factor, and $r_{u(g,psych)}$ is the residual genetic covariance across g and the psychiatric factor. The denominator of this equation then reflects the total genetic covariance (r_g) across g and the psychiatric factor.”

We note that although the reviewer’s exact suggestion to incorporate the external correlates into the model as additional phenotypes of g is statistically possible it stands in contrast to our own take on the framework that we employ and the relationship between cognitive phenotypes and these disease traits. More specifically, we do not take the external disease correlates to be part of the same g -factor construct defined by sets of cognitive phenotypes. Put another way, although the g -factor is not a tangible entity, the modeling of genetic g reflects a theoretical account of shared biological processes that give rise to pervasive patterns of overlap across cognitive traits. It is our sense that these new analyses that are structured to include the external correlates as separable constructs both answer the reviewer’s research question and align with theoretical accounts of the underlying processes.

7. They do a gene-set analysis based on the detected genes with both the g-factor and Q_Gene associations to identify pathways, and also perform pathway enrichment by stratified Genomic SEM. It is not quite clear how consistent of the two sets of results based on T-SEM and Genomic SEM analysis.

We agree that this was not made clear before and have added text to the Introduction to clear this up at the start. We specifically write on p. 3:

“T-SEM and Stratified Genomic SEM are distinguishable with respect to the biological substrate being examined—tissue-specific gene expression versus categories of genes, respectively—but are both applied here with the shared end goal of elucidating the functional biology that is common and unique across cognitive domains.”

As we note in the above text, while the level of analysis is distinct across T-SEM and Stratified Genomic SEM, we believe that the inclusion of both sets of results within the same manuscript follows a field standard in complex trait genomics to apply multiple methods to elucidate many aspects of an overarching research question or topic.

8. Is it possible to write/explain their model in the form of linear mixed model? I think it would be helpful for readers to better understand their method if they can link the factor analysis to linear models as many researchers in the field are more familiar with the latter.

We agree that this would be useful for different researchers and have updated the Online Method to include the linear equations of the measurement and structural model within the general T-SEM framework, and for the current analyses applied to examining gene expression effects on genetic g. We now write on ps. 10-11:

“In Stage 2, a model is specified in which gene expression is associated with some other parameter in the model, such as a latent factor defined by the genetic components of the included phenotypes. The model itself can be broken into two parts. The first reflects the measurement model, which parsimoniously describes the genetic relationships across k analyzed traits via a smaller subset of m latent variables. This can be expressed as:

$$Y_g = \Lambda\eta + U$$

where Y_g is a k -length vector of the genetic component of the analyzed traits, η is an m -length vector of latent variables, Λ is a $k \times m$ matrix of factor loadings, and U is a k -length vector of residual genetic variances not accounted for by the latent variables.

In T-SEM, the structural model is then added on top of the genomic measurement model in order to relate tissue-specific gene expression to the latent variables, and the latent variables to one another when > 1 latent variables are estimated. The structural model in T-SEM can be expressed as:

$$\eta = B\eta + \Gamma x + E,$$

where η is again an m -length vector of latent variables, B is an $m \times m$ matrix of regression coefficients that relate latent variables to one another, Γ is an m -length vector of regression coefficients relating the latent variables to tissue-specific gene expression, x is the tissue-specific gene expression, and E is an m -length vector of the residual variances of the latent variables.

In the context of the current analyses, the g -factor measurement model can be written according to the following system of linear equations:

$$\begin{bmatrix} v_{g\text{Matrices}} \\ v_{g\text{Memory}} \\ v_{g\text{RT}} \\ v_{g\text{SD}} \\ v_{g\text{Trails-B}} \\ v_{g\text{Tower}} \\ v_{g\text{VNR}} \end{bmatrix} = \begin{bmatrix} \lambda_{\text{Matrices}} \\ \lambda_{\text{Memory}} \\ \lambda_{\text{RT}} \\ \lambda_{\text{SD}} \\ \lambda_{\text{Trails-B}} \\ \lambda_{\text{Tower}} \\ \lambda_{\text{VNR}} \end{bmatrix} g + \begin{bmatrix} u_{\text{Matrices}} \\ u_{\text{Memory}} \\ u_{\text{RT}} \\ u_{\text{SD}} \\ u_{\text{Trails-B}} \\ u_{\text{Tower}} \\ u_{\text{VNR}} \end{bmatrix},$$

where v_g reflects the genetic component of each of the seven cognitive phenotypes, the λ 's are the phenotype-specific factor loadings on g , and the u 's denote the residual genetic variances of the phenotypes. The effect of tissue-specific gene expression on g can then be expressed as:

$$g = \gamma x + e,$$

where γ is the the unstandardized regression coefficient of tissue-specific gene expression on g , x is the tissue-specific gene expression, and e the residual variance of g ."

Reviewer #2 (Remarks to the Author):

This paper is to develop and validate Transcriptome-wide Structural Equation Modeling (T-SEM), a novel method for studying the effects of tissue-specific gene expression within a multivariate space. Although it has several interesting findings, this reviewer has several major concerns about this paper.

1. The T-SEM can be regarded as a straightforward extension of Genomic SEM (Grotzinger et al., 2019, Nature Human Behavior). The only difference is to replace SNP with Gene in the GSEM. The estimation procedure is the standard two-stage WLS procedure. Therefore, from statistical perspective, the contributions are at most marginal.

We respect this reviewer's viewpoint, but disagree for reasons we aim to detail here. First, we note that great care was taken to ensure that the statistical framework employed by Genomic SEM is well calibrated when including gene expression effects (something that we do not take to be given). This includes conducting an extensive set of simulations at the gene expression level to validate the properties of T-SEM, including establishing appropriate Type I error control and power. Second, T-SEM is able to guard against false positive inference that would otherwise occur, such as when summary statistics from multivariate GWAS are submitted to univariate TWAS. Indeed, in response to Reviewer 1's comments, we have also updated the manuscript to include a second set of simulations illustrating this important point.

Third, we highlight that integrating gene-expression analyses into the Genomic SEM package required extensive updates, including a separate function for reading in univariate TWAS results, creating a pipeline for incorporating the effect of gene expression within a multivariate system of genetically correlated traits, and carefully documenting on the Genomic SEM GitHub how to utilize these package updates. Without these updates, the original Genomic SEM package could not be used to perform the analyses described here. As such, it is incorrect that the methods and results reported in this submission represent the rote application of previously developed statistical and computational methods.

2. Most references have not been cited appropriately throughout the paper. To name a few, for instance, the reference for GSEM should be [20], whereas the authors cited [21] on page 9. Please see “T-SEM estimation follows the general two-stage approach introduced in the Genomic SEM framework,²¹”. The reference [21] is not about [20]. This reviewer checked all major references, including the reference for FUSION, and found that most references are incorrectly cited.

We apologize for this inconsistency and thank the reviewer for pointing out this error. The references have been corrected throughout the manuscript.

3. Although the authors gave some details about T-SEM, many key steps strongly depend other methods so that this reviewer has to read more than 10+ papers to understand the proposed methods. Moreover, stratified Genomic SEM does not have details so that readers could hardly understand.

We thank the reviewer for pointing this out and agree that the original submission lacked sufficient background details on each of the methods. We have endeavored to provide additional details of each method throughout the updated manuscript, including the addition of the system of equations employed by T-SEM in response to Reviewer 1 Comment 8. We also now include an expanded overview of T-SEM, and TWAS within FUSION, on p. 4 of the main text:

“Overview of T-SEM

T-SEM is a novel method for examining the effect of tissue-specific gene expression on any parameter within the general Genomic Structural Equation Modelling (Genomic SEM) framework.¹² T-SEM follows a two-stage approach. In Stage 1, univariate, summary-based TWAS is estimated for each of the included traits in order to perform summary-based transcriptomic imputation (TI), which estimate the effects of tissue-specific gene expression on individual GWAS phenotypes. The analytic pipeline outlined here, and the corresponding open-source publicly available software, specifically utilizes summary-based TWAS output from the FUSION software.¹³ Summary-based TWAS is estimated in FUSION as the sum of GWAS Z-statistics weighted by what is referred to as the functional weights. These functional weights are typically pre-compiled from smaller reference datasets containing both tissue-specific gene expression and genotype data, and can generally be described as indexing the association between individual single nucleotide polymorphisms (SNPs) and gene expression. Given the costly and intensive nature of obtaining gene expression data, particularly from tissue types such as specific brain regions, summary-based TWAS then allows for drawing inferences about patterns of gene expression associated with complex traits for which only GWAS summary statistics are available. Summary data from each univariate TWAS produced by FUSION are then combined with one another and with the empirical genetic covariance matrix produced using the multivariable version of LDSC¹⁴ within Genomic SEM to produce a complete genetic covariance matrix for expression of each gene and the GWAS phenotypes (S_{Full}).¹² An associated sampling covariance (V_{SFull}) matrix is also constructed. V_{SFull} includes squared standard errors (SEs) on the diagonal, and sampling covariances on the off-diagonal that quantify dependencies between sampling errors of the estimates. These off-diagonal elements allow T-SEM to be performed for traits with unknown levels of participant overlap across the contributing GWAS.

In Stage 2, the user specifies a SEM in which gene expression is associated with the multivariate system of heritable phenotypes via regression or covariance relationships with components of the model. In our empirical application, the SEM consists of a general factor indexing genetic overlap across seven cognitive traits. We also produce a Q_{Gene} statistic that indexes the extent to which there is violation of the

null hypothesis that imputed expression of a given gene affects the individual traits strictly via the factor. Larger Q_{Gene} statistics occur when gene expression is highly specific to an individual trait or when the gene expression effect is directionally opposing across traits. Thus, we use T-SEM to identify genes whose tissue-specific expression has general effects on a system of diverse cognitive traits and distill them from genes with more trait-specific effects.”

With respect to Stratified Genomic SEM we now write on p. 7 of the main text:

“Stratified Genomic SEM Analysis of Cognitive Traits

Stratified LDSC²² is a method for partitioning the SNP-based heritability for a different trait across different classes of genetic variants, referred to as functional annotations, that are grouped according to some shared characteristic. These shared characteristics can include, for example, whether the variants tend to be conserved in mammals, are associated with specific histone marks, or are implicated in neuronal subtypes. A functional annotation is then considered to be enriched, indicating that it is of particular relevance for a given trait, when the genetic variance within that annotation is greater than the proportional size of that annotation. The proportional size of the annotation reflects the number of SNPs in the annotation over the total number of SNPs analyzed. Stratified Genomic SEM is a recently developed, multivariate corollary of S-LDSC that can be used to examine enrichment of any model parameter estimated in Genomic SEM, thereby allowing for examining functional enrichment within a multivariate space.¹¹ This facilitates identifying categories for which pleiotropic variation, as separable from trait-specific genetic variation, is enriched.”

4. The whole paper is not well written. The logic does not flow very well. For instance, the second paragraph does not connect with the first and third paragraphs very well.

We have made updates throughout the manuscript to improve readability. With respect to the Introduction section pointed out by the reviewer, we have updated paragraph 2 to be conceptually tied to the surrounding paragraphs and now write on p. 3:

“The finding that many diverse cognitive functions are positively intercorrelated was discovered by Spearman in 1904, and has come to be known as “arguably the most replicated result in all psychology.”¹ Spearman speculated that the positive manifold of test intercorrelations resulted from their mutual reliance on a common factor, which he termed general intelligence, or g , but that each cognitive test also relied on more-specific factors, s . Spearman developed factor analysis in order to estimate the relative contributions of g and s factors to a given test from empirical data. Current evidence indicates that g accounts for between approximately 40% and 60%² of variation in cognitive test scores. In addition, the g -factor is associated with a range of important life outcomes including income level,³ educational attainment,⁴ social mobility,⁵ health,⁶ and longevity.⁷ However, despite over 100 years of debate regarding the nature and mechanisms of g , the biological mechanisms shared across cognitive functions have remained relatively elusive.

Family-based designs have long indicated that genetic sharing across cognitive functions may underly Spearman’s positive manifold.⁸ More recently, genome-wide association studies (GWAS) have identified specific genetic variants that are associated with the genetic overlap across diverse cognitive traits.⁹ A clear next step in this line of research is to characterize the biological pathways implied by these more recent GWAS results. Functional genomic approaches parsimoniously distill the genetic

signal across millions of genetic variants into biologically meaningful mechanisms. For example, a recent study of educational attainment (EA) determined that associated genetic variants were enriched for genes involved in specific neurophysiological functions, including synaptic plasticity, ion channel activation, and neurotransmitter secretion.¹⁰ However, as we demonstrate via simulation, existing functional genomic approaches have been developed for univariate applications and are ill-equipped to analyze multivariate genomic data without false positive inference.

The current study performs the first, multivariate functional genomic analysis of g to both leverage the shared power across seven cognitive traits for novel discovery and elucidate the biological pathways unique to, and shared across, these traits. We specifically apply two novel multivariate methods. First, we introduce and validate Transcriptome-wide Structural Equation Modeling (T-SEM), a method that extends Transcriptome-Wide Association Studies (TWAS) approaches to estimate the effects of tissue-specific gene expression within a multivariate system of GWAS traits. Using data from UK Biobank, we apply T-SEM to estimate relationships between gene expression and g in order to identify biological mechanisms of sharing across seven cognitive traits. We validate and employ a heterogeneity statistic (Q_{Gene}) within T-SEM that quantifies the extent to which the data deviate from the hypothesis that gene expression affects the traits strictly via a common factor, such as g . This allows us to identify tissue specific patterns of gene expression that are associated with only a subset of cognitive traits, or one cognitive trait, such as reaction time. In order to understand broader biological pathways that transcend expression of individual genes, we go on to apply another recently developed multivariate functional method, Stratified Genomic SEM¹¹ to examine genetic sharing and uniqueness within different classes of genetic variants (e.g., variants associated with specific neuronal subtypes). T-SEM and Stratified Genomic SEM are distinguishable with respect to the biological substrate being examined—tissue-specific gene expression versus categories of genes, respectively—but are both applied here with the shared end goal of elucidating the biology that is common and unique across cognitive domains.”

While we certainly do not view it as the reviewer’s job to copy-edit the entirety of the manuscript, if there are specific sections that they feel in particular require additional revisions we are happy to do so.

Review References

- Andersen, M. S., Bandres-Ciga, S., Reynolds, R. H., Hardy, J., Ryten, M., Krohn, L., ... & International Parkinson's Disease Genomics Consortium. (2021). Heritability enrichment implicates microglia in Parkinson's disease pathogenesis. *Annals of Neurology*, *89*(5), 942-951.
- Hill, W. D., Davies, G., Harris, S. E., Hagenaars, S. P., Liewald, D. C., Penke, L., ... & Deary, I. J. (2016). Molecular genetic aetiology of general cognitive function is enriched in evolutionarily conserved regions. *Translational psychiatry*, *6*(12), e980-e980
- Li, Y. I., Wong, G., Humphrey, J., & Raj, T. (2019). Prioritizing Parkinson's disease genes using population-scale transcriptomic data. *Nature communications*, *10*(1), 1-10.
- Liao, C., Laporte, A. D., Spiegelman, D., Akçimen, F., Joobar, R., Dion, P. A., & Rouleau, G. A. (2019). Transcriptome-wide association study of attention deficit hyperactivity disorder identifies associated genes and phenotypes. *Nature communications*, *10*(1), 1-7.
- Zhao, B., Shan, Y., Yang, Y., Yu, Z., Li, T., Wang, X., ... & Zhu, H. (2021). Transcriptome-wide association analysis of brain structures yields insights into pleiotropy with complex neuropsychiatric traits. *Nature communications*, *12*(1), 1-11.

REVIEWER COMMENTS

Reviewer #1 (Remarks to the Author):

I would like to thank the authors' effort to address my comments in detail. While I think most of them have been adequately addressed, I have a follow-up question regarding to their response to my previous comment #1.

The authors stated the importance of Q_Gene , which guards against false positives and identifies sources of differentiation across genetically correlated traits. I don't quite follow why only T-SEM can guard against false positives otherwise cannot. The heterogeneity of gene effects comes from the heterogeneity of genetic effects. Since genomic SEM provides Q_SNP for heterogeneity of genetic effects, the same goal can be achieved by filtering variants with large Q_SNP and then performing TWAS. I believe this should be explicitly tested in their simulations whether this approach would cause inflation in false positives or result in a lower power than Q_Gene from T-SEM. Their results in the new simulations show that the power for detecting common factors are almost the same between the two approaches (Supplementary Fig. 6). There remains seem to be a lack of strong support for the novelty of the method without a clear demonstration of the benefit in detecting the heterogeneity.

Reviewer #3 (Remarks to the Author):

Review: Transcriptome-wide and Stratified Genomic Structural Equation Modeling Identify Neurobiological Pathways Shared across Diverse Cognitive Traits

Overview: Grotzinger et al propose the transcriptome-wide structural equation modeling (T-SEM) framework, which aggregates eQTL results from reference panels with genome-wide association data to identify genes associated with a latent parameter g , which aims to capture a notion of general intelligence. The statistical model is largely the result of extending the authors previous multivariable LDSC/Genomic-SEM framework to account for the weighted linear combination of GWAS data under eQTL weights inferred using FUSION. The authors then propose a heterogeneity statistic, Q_gene , which they use to quantify the extent to which genetically regulated expression associates affects downstream traits outside of the latent parameter g . The authors validate their approach using simulations, and then apply it to GWAS summary data from 7 cognitive traits.

Major Comments:

1. The authors have included a description of the model using the linear mixed model framework, which is helpful, but I find the overall presentation to have some errors which makes it somewhat difficult to follow. For example, η shows up on both left/right sides of an equation, this is likely a typo, or the result of some overloaded η , but regardless the presentation should be a bit more polished.

2. The authors have performed a significant number of simulations, and for that I commend them, however in all scenarios the effects of genetics on gene expression were fixed and known a-priori. What would be helpful would be to see results under simulations where there exist true generative eQTL effects, but what researchers have in hand are either noisy estimates, or are noisy and at best correlated estimates (ie proxy context). These results are likely to attenuate any findings, but quantifying this effect will help place downstream results in context.

Reviewer #1 (Remarks to the Author):

I would like to thank the authors' effort to address my comments in detail. While I think most of them have been adequately addressed, I have a follow-up question regarding to their response to my previous comment #1.

The authors stated the importance of Q_{Gene} , which guards against false positives and identifies sources of differentiation across genetically correlated traits. I don't quite follow why only T-SEM can guard against false positives otherwise cannot. The heterogeneity of gene effects comes from the heterogeneity of genetic effects. Since genomic SEM provides Q_{SNP} for heterogeneity of genetic effects, the same goal can be achieved by filtering variants with large Q_{SNP} and then performing TWAS. I believe this should be explicitly tested in their simulations whether this approach would cause inflation in false positives or result in a lower power than Q_{Gene} from T-SEM. Their results in the new simulations show that the power for detecting common factors are almost the same between the two approaches (Supplementary Fig. 6). There remains seem to be a lack of strong support for the novelty of the method without a clear demonstration of the benefit in detecting the heterogeneity.

We agree that further justification for T-SEM over TWAS of summary statistics from a common factor GWAS was needed. To this end, we compare results obtained from TWAS of GWAS summary statistics from a common factor GWAS of the g-factor excluding genes with Q_{SNP} hits to estimates obtained directly from T-SEM. Note that we employ a Q_{SNP} filtering procedure of removing any gene that has functional weights for a Q_{SNP} variant, where Q_{SNP} variants were themselves defined using the T-SEM Bonferroni corrected p-value threshold, as opposed to the more stringent genome-wide significance threshold. Both of these decisions err on the liberal side of excluding more Q_{SNP} signal so as to prevent this heterogeneous signal from contaminating the subsequent TWAS. Even still, we find that TWAS results filtered on Q_{SNP} continues to identify 2 gene hits that T-SEM finds to be significant for Q_{Gene} .

Pragmatically, it is of note that this specific Q_{SNP} filtering process would be quite cumbersome in-practice for users as it requires individually loading in each chromosome- and tissue-specific weights file to cross-reference with Q_{SNP} hits. We highlight here that the more straightforward process of filtering the GWAS summary statistics on Q_{SNP} prior to running TWAS did not remove any of the Q_{Gene} hits. This is both because FUSION performs imputation of missing GWAS Z-statistics when possible, which is not a feature that the user is given the option of turning off, and because genes are defined by as many as 2,038 variants with functional weights. These real data findings coupled with our existing simulations indicate that T-SEM is uniquely suited to identify heterogeneity in patterns of gene expression. As genetic divergence is itself a central research question in the field, and only possible to test with T-SEM, we view the validation and application of T-SEM to be an important and novel contribution. The text has been updated on p. 7 of the Results section:

"We went on to compare results obtained from T-SEM to results for a TWAS of the g-factor GWAS summary statistics. Consistent with simulation results, TWAS and T-SEM estimates for g were highly correlated ($r > .99$; Supplementary Figure 14). For the TWAS of the g-factor GWAS summary statistics we then employed a Q_{SNP} filtering procedure of removing any gene that had functional weights for a Q_{SNP} variant, where Q_{SNP} variants were themselves defined using the T-SEM Bonferroni corrected p-value threshold, as opposed to the more stringent genome-wide significance threshold. Both of these decisions err on the lax side of excluding more Q_{SNP} signal so as to prevent such signal from contaminating the subsequent TWAS. This Q_{SNP} filtering process failed to remove 23 of the 156 Q_{Gene} hits identified by T-SEM. Moreover, 2 of the 23 remaining Q_{Gene} hits were identified as TWAS hits for g and, given significant

Q_{Gene} findings, likely to be false positives (Supplementary Figure 14).¹ Thus, even under conditions selected to ensure that heterogeneous SNP effects were removed, TWAS of the g-factor GWAS summary statistics was less effective than T-SEM at pruning out heterogeneous signal. In summary, findings from both simulations and application to real data indicate that T-SEM is uniquely suited to guard against false positives for effects of gene expression on general factors and to identify patterns of gene expression that underly genetic divergence among correlated phenotypes.”

Reviewer #3 (Remarks to the Author):

Review: Transcriptome-wide and Stratified Genomic Structural Equation Modeling Identify Neurobiological Pathways Shared across Diverse Cognitive Traits

Overview: Grotzinger et al propose the transcriptome-wide structural equation modeling (T-SEM) framework, which aggregates eQTL results from reference panels with genome-wide association data to identify genes associated with a latent parameter g , which aims to capture a notion of general intelligence. The statistical model is largely the result of extending the authors previous multivariable LDSC/Genomic-SEM framework to account for the weighted linear combination of GWAS data under eQTL weights inferred using FUSION. The authors then propose a heterogeneity statistic, Q_{gene} , which they use to quantify the extent to which genetically regulated expression associates affects downstream traits outside of the latent parameter g . The authors validate their approach using simulations, and then apply it to GWAS summary data from 7 cognitive traits.

Major Comments:

1. The authors have included a description of the model using the linear mixed model framework, which is helpful, but I find the overall presentation to have some errors which makes it somewhat difficult to follow. For example, η shows up on both left/right sides of an equation, this is likely a typo, or the result of some overloaded η , but regardless the presentation should be a bit more polished.

The η on both sides of the equation appears counter-intuitive but is a way to represent a system of equations where all variables have the potential to be both predictors and outcomes. This is because the B matrix contains a set of a parameters that reflect the model specified, wherein some parameters are freely estimated and others are fixed to 0. This matrix can consequently be thought of as a “mask matrix” that prevents an overloaded η by fixing certain parameters, such as the latent variable predicting itself, to 0. We agree that this should be made more explicit as other readers are likely to have the same question. To this end, we have updated the noted section to read (p. 12):

“The structural model in T-SEM can be expressed as:

$$\eta = B\eta + \Gamma x + E,$$

where η is again an m -length vector of latent variables, B is an $m \times m$ matrix of regression coefficients that relate latent variables to one another, Γ is an m -length vector of regression coefficients relating the latent variables to tissue-specific gene expression, x is the tissue-specific gene expression, and E is an m -length vector of the residual variances of the latent variables. The terms in B , Γ , and E may include both free parameters and fixed terms, as specified by the user to represent the model of interest. We note that η appears on both the left and right side of the equation as we utilize all-y notation, which does not distinguish between endogenous and exogenous latent variables.³⁸ The B matrix of regression coefficients

then prevents specifying the regression of a latent variable predicting itself by fixing those specific parameters to 0 for that cell of the matrix.”

2. The authors have performed a significant number of simulations, and for that I commend them, however in all scenarios the effects of genetics on gene expression were fixed and known a-priori. What would be helpful would be to see results under simulations where there exist true generative eQTL effects, but what researchers have in hand are either noisy estimates, or are noisy and at best correlated estimates (ie proxy context). These results are likely to attenuate any findings, but quantifying this effect will help place downstream results in context.

We thank the reviewer for pointing out this important point and have added a new set of simulations that explicitly take into account eQTL effects that reflect only a proxy of the population estimate. We find, as would be expected, that this increases the variation in estimates. We agree that this is a useful contextual consideration for interpreting our results. We now write on p. 18:

“We ran an additional set of simulations that examined the effect of sampling variation in estimates of the SNP effect on gene expression (b_{eQTL}). The sampling variance (i.e., the squared standard error) of the b_{eQTL} estimate is specifically given as:

$$\sigma_{beQTL}^2 = (SE_{beQTL})^2 = \frac{1 - \sigma_{SNP}^2 (b_{eQTL}^2)}{\sigma_{SNP}^2 (n-1)}.$$

For Scenario 1, where the generating population matched the factor model, we then generated a set of 100 population b_{eQTL} estimates using the `rnorm` function in R. These were subsequently used to create a new set of population b_{GWAS} estimates for each simulation run, the SNP effects (b_{GWAS}) sampled from the sampling distribution given above, and the remainder of the simulation pipeline conducted to mirror the other SNP-level simulations. Importantly, the FUSION weight for GNL3 was left unchanged, such that each simulation varied in the degree of mismatch between the population b_{eQTL} and the functional weight used to produce TWAS estimates.”

We continue on p.19 to discuss these simulation results:

“Simulation results that included sampling variation in the b_{eQTL} estimates are visually summarized in Supplementary Figure 8. We find that, as with the other simulation findings, that there was a tight correspondence between T-SEM estimates and those from $TWAS_{Factor}$, that these are both well-powered approaches when the population matches the factor model with 93% of the simulation runs significant at a Bonferroni corrected threshold, and that Q_{Gene} evinces a well-controlled FPR of 5% at $p < .05$. Finally, as would be expected, the downstream consequence of including variation in the population b_{eQTL} estimates was a wider sampling distribution of estimates relative to the simulations that treated b_{eQTL} as a known value (Supplementary Table 3). As with univariate TWAS, functional weights estimated from finite samples will result in greater variation in estimates relative to the population. As the gene expression samples used to train these functional weights increase, imprecision of the TWAS weights will exert an increasingly minimal influence on the precision of downstream estimates.”

REVIEWERS' COMMENTS

Reviewer #1 (Remarks to the Author):

The authors have adequately addressed my comment.

Reviewer #3 (Remarks to the Author):

The authors have addressed my previous comments.